# MV-RAG: Retrieval Augmented Multiview Diffusion

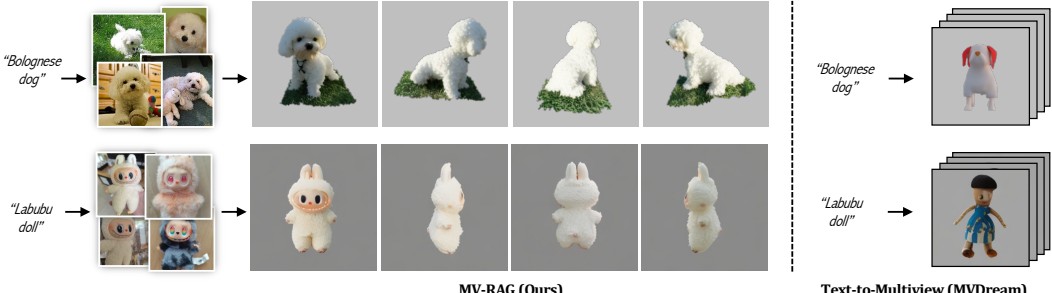

Figure 1: We introduce a retrieval-augmented diffusion framework for text-to-multiview generation. Given a text prompt, our method retrieves real-world images and adaptively leverages them together with the text, enabling faithful generation of out-of-distribution and newly emerging objects.

## ABSTRACT

Text-to-3D generation approaches have advanced significantly, producing high-quality and 3D-consistent outputs. However, they often fail to produce out-of-domain (OOD) or rare concepts, yielding inconsistent or inaccurate results. To this end, we propose MV-RAG, a novel text-to-3D pipeline that first retrieves relevant 2D images from a large in-the-wild 2D database and then conditions a multiview diffusion model on these images to synthesize consistent and accurate multiview outputs. Training such a retrieval-conditioned model is achieved via a novel hybrid strategy bridging structured multiview data and diverse 2D image collections. This involves training on multiview data using augmented conditioning views that simulate retrieval variance for view-specific reconstruction, alongside training on sets of retrieved real-world 2D images using a distinctive held-out view prediction objective: the model predicts the held-out view from the other views to infer 3D consistency from 2D data. We also introduce a prior-guided fusion mechanism that dynamically balances retrieval signals with the model's prior. To facilitate a rigorous OOD evaluation, we introduce a new collection of challenging OOD prompts. Experiments against state-of-the-art text-to-3D, image-to-3D, and personalization baselines show that our approach significantly improves 3D consistency, photorealism, and text adherence for OOD/rare concepts, while maintaining competitive performance on standard benchmarks.

## 1 INTRODUCTION

The automated generation of 3D content from text is important for applications such as game modeling, computer animation and virtual reality. Current approaches largely leverage pre-trained 2D text-to-image diffusion models Song et al. (2021); Ho et al. (2020) as visual and semantic priors, either via optimization or to train generative models that produce consistent multiview images. These methods yield high-quality outputs. However, they often struggle with out-of-domain (OOD) or rare prompts, producing geometrically inconsistent results (e.g., poorly rendered unseen regions) or failing to adhere to the text, hallucinating details or replacing rare concepts with common ones.

A common text-to-3D approach uses Score Distillation Sampling (SDS) Poole et al. (2023); Lin et al. (2023); Liang et al. (2024) to optimize a 3D representation such as NeRF Mildenhall et al. (2020)

by distilling knowledge from 2D text-to-image models. While high-fidelity, SDS-based methods often inherit the 2D prior's limitations on OOD prompts, yielding flawed 3D assets. To address this, recent work Seo et al. (2024); Chen et al. (2024) explores retrieval augmentation, incorporating existing 3D assets as geometric priors. This improves consistency for database concepts but remains limited by scale and diversity. 3D personalization techniques, e.g., DreamBooth3D Raj et al. (2023), adapt a pretrained 2D model to a specific subject using a few (3-6) images, followed by SDS. These methods capture subject-specific details but require inference-time fine-tuning and still face geometry inconsistencies inherent to SDS. Feed-forward multiview diffusion models Shi et al. (2023b); Liu et al. (2023a); Long et al. (2024) synthesize consistent multiview images from text or images, often fine-tuned on large 3D datasets like Objaverse Deitke et al. (2023b). This enhances 3D awareness and improves geometric consistency over 2D-lifting approaches. Still, they struggle with OOD or rare concepts due to limited coverage in both 2D priors and 3D fine-tuning data, producing outputs with reduced photorealism, inconsistent views, or poorly inferred unobserved regions.

To this end, we propose MV-RAG, a multiview diffusion model that conditions generation on relevant in-the-wild unposed 2D images retrieved from large-scale collections. By leveraging retrieval, MV-RAG produces consistent multiview images even for rare or out-of-distribution (OOD) concepts. To enable conditioning on varying, in-the-wild views, our training combines supervision from two novel sources: (1). a *reconstruction objective* on structured multiview data, where we use augmented, "retrieval-like" conditioning views to enforce robust geometric consistency; and (2). a *hold-one-out objective* on 2D image-text data, where the model learns to infer 3D relationships by generating a held-out image from a set of $K$ related views. This hybrid scheme enables MV-RAG to learn 3D coherence while generalizing with the diverse appearance knowledge from 2D priors. To balance the influence of the base model's prior with external retrieval signals, we further introduce a fusion mechanism that dynamically adapts to the OODness of the prompt. We note that adapting RAG to the multiview setting is non-trivial, posing two key challenges requiring 3D reasoning, which are not required in the single-image RAG setting: (1) ensuring cross-view consistency when retrieved images offer only partial views, and (2) coherently composing features from different retrieved images (e.g., a car's grille from one, its wheels from another) into a single object. Our hybrid training scheme is designed to address both (see further discussion in Appendix A.3).

To evaluate OOD scenarios, we curate a new benchmark of 196 challenging prompts paired with retrieved images. On this benchmark, MV-RAG significantly outperforms text-to-3D, image-to-3D, and personalization baselines in terms of 3D consistency, photorealism, and text alignment, while remaining competitive on standard in-domain benchmarks. Ablation studies further validate our design choices. An overview of our method is shown in Fig. 1.

**Contributions.** We make the following contributions: **(1)**. The first framework to successfully apply retrieval-augmented generation (RAG) to multiview 3D synthesis, achieving superior performance on out-of-distribution (OOD) concepts. **(2)**. A novel hybrid 2D-3D training scheme that bridges the gap between structured 3D data and unposed 2D image collections. **(3)**. A novel prior-guided attention mechanism that dynamically balances the model's internal prior with external retrieval signals. **(4)**. **OOD-Eval**, a new benchmark of challenging prompts to facilitate research on OOD 3D generation.

## 2 RELATED WORK

**3D Generation Using 2D Diffusion Models** Generating 3D content by leveraging strong priors from 2D diffusion models Ho et al. (2020) is a dominant paradigm. One major approach optimizes 3D representations, such as Neural Radiance Fields (NeRFs) Mildenhall et al. (2020) and more recently 3D Gaussian Splatting (3DGS) Kerbl et al. (2023), via Score Distillation Sampling (SDS) Poole et al. (2023); Lin et al. (2023); Tang et al. (2024b), directly distilling knowledge from 2D priors. However, SDS often struggles with geometric consistency and fidelity due to weak 3D awareness in the priors Hong et al. (2023); Shi et al. (2023b). Feed-forward multi-view diffusion models Shi et al. (2023b); Huang et al. (2024), often fine-tuning 2D diffusion priors with 3D dataset supervision, enhance geometric stability by directly generating multiple consistent views Shi et al. (2023b). However, these models struggle with out-of-domain (OOD) or rare concepts due to limitations in 2D priors and insufficient 3D training data coverage, leading to reduced photorealism or inconsistent geometry. Related image-to-multiview approaches Liu et al. (2023a); Shi et al. (2023a); Wang & Shi (2023); Liu et al. (2023c); Long et al. (2024), while effective with single clear inputs, are ill-suited

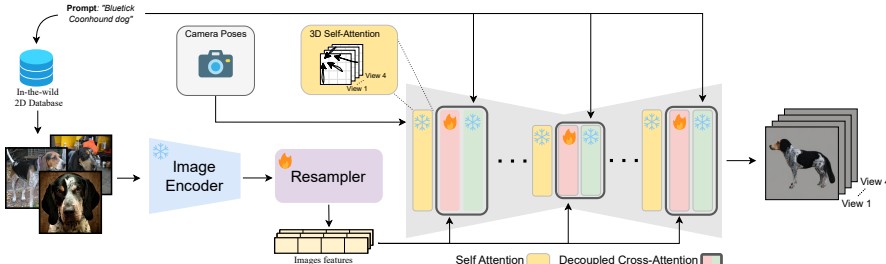

Figure 2: **Overview of our pipeline.** Given a text prompt, we retrieve $k$ relevant images from an in-the-wild 2D image corpus. Local features are extracted from each image, projected through a Resampler and integrated into retrieval-attention modules to guide the multi-view generation process.

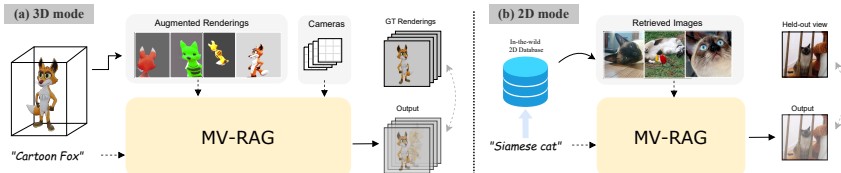

Figure 3: **Overview of our training scheme.** We adopt a hybrid training strategy that alternates between two modes. **3D mode (LHS):** A 3D object is rendered to produce ground-truth multi-view images. Additional views are generated and subjected to augmentations to serve as retrievals. These augmented views, along with the target camera parameters and the associated prompt, are provided as input to the model. **2D mode (RHS):** We retrieve $K + 1$ images from a 2D corpus, where $K$ images are used as retrievals and one held-out image serves as the target view. In this mode, the model performs 2D self-attention rather than 3D attention, and no target camera parameters are provided.

for leveraging multiple, varied, unposed retrieved images. Our work builds on these advancements, targets OOD generation by training a multiview diffusion model to incorporate retrieved 2D images.

**Retrieval Augmented Generation (RAG)**    RAG improves generative models by incorporating external information, aiding the handling of OOD/rare entities without retraining, a successful paradigm in NLP Lewis et al. (2020); Borgeaud et al. (2022). Notably, Soudani et al. (2024) show that RAG is preferable to fine-tuning, especially for OOD/rare concepts. In 2D image synthesis, RAG methods similarly use retrieved images or text pairs to enhance fidelity for uncommon concepts or guide generation Chen et al. (2022); Sheynin et al. (2022); Blattmann et al. (2022); Shalev-Arkushin et al. (2025). Recently, text-to-3D generation methods like RetDream Seo et al. (2024) and Sculpt3D Chen et al. (2024) retrieve existing *3D assets* for geometric priors to improve optimization consistency. However, this is limited by the scarcity and diversity of 3D databases, especially for OOD or rare concepts. Inspired by findings in NLP, our work performs RAG in multiview generation via MV-RAG, leveraging abundant *2D image datasets* to condition a multiview diffusion model, offering a scalable way to ground OOD concept generation in real-world data.

**Personalization.**    Personalization methods adapt generative models to specific subjects from only a few examples, either by optimizing embeddings or fine-tuning the model Gal et al. (2023); Ruiz et al. (2023). These ideas have been extended to 3D: DreamBooth3D Raj et al. (2023) combines SDS optimization with a subject-specific 2D prior, while multiview diffusion models such as MVDream Shi et al. (2023b) can be fine-tuned in a DreamBooth-like fashion. Such approaches typically require per-subject fine-tuning or inference-time optimization. By contrast, we integrate retrieved 2D image sets representing general concepts directly into MV-RAG's training. Furthermore, unlike personalization which assumes all inputs depict the same instance, our method is designed to compose features from multiple, potentially different instances of a category. This enables the model to handle diverse, especially OOD, concepts without subject-specific adaptation at inference.

## 3 METHOD

Given an input text prompt $p$, we first retrieve relevant 2D images corresponding to $p$ from a corpus of 2D text-image pairs. Then, we use these images, along with $p$, to guide the generation process of a multi-view diffusion model. An overview of our approach is provided in Fig. 2. We begin

by describing the training process, which involves a data preprocessing stage to prepare distinct conditioning and target data for 2D or 3D training modes, followed by the training process.

## 3.1 Training Data Preprocessing

Our model training leverages geometric grounding from 3D datasets and diversity from 2D datasets. For both, we prepare 2D conditioning images related to a text $p$, simulating inference-time retrieval, alongside target supervision data. Fig. 3 illustrates our separate 2D and 3D training modes.

**2D Data Mode Supervision.** Here, we utilize a large-scale 2D in-the-wild text-image dataset (images are neither posed nor aligned). For a text prompt $p_i$, we consider $K + 1$ relevant images, from which we designate $K$ as the conditioning retrieved views $\mathcal{I}_{\text{ret}} = \{I_i\}_{i=1}^{K}$, and the remaining image, $I_{\text{target}}$, serves as the target image on which the diffusion loss is computed. This process yields training samples of the form $\mathcal{D}_{2D} = \{p, \mathcal{I}_{\text{ret}}, I_{\text{target}}\}$. No ground truth camera poses are assumed.

**3D Data Mode Supervision.** For 3D data, we assume a dataset comprising text prompts and corresponding 3D object models. For each 3D object, we render a set of $N$ ground truth target views $I_{\text{target}} = \{I_i\}_{i=1}^{N}$ at $N$ camera poses $\mathcal{C}$. We follow MVDream and use 4 orthogonal camera poses. To simulate the diverse nature of retrieved conditioning images, we render the object from $K$ additional random poses followed by a sequence of random augmentations. This yields conditioning views $\mathcal{I}_{\text{ret}} = \{I_i\}_{i=1}^{K}$. We apply a combination of geometric and semantic augmentations designed to mimic in-the-wild variability and enhance generalization. Crucially, these simulated retrievals are treated as unposed, and no camera information is provided for them during training. This yields training samples of the form $\mathcal{D}_{3D} = \{p, \mathcal{C}, \mathcal{I}_{\text{ret}}, \mathcal{I}_{\text{target}}\}$. See Appendix Sec.A.6 for additional details. We note that training by conditioning on real-world 2D retrievals was suboptimal, as the retrieved instances often differed significantly from the ground-truth 3D object, creating a conflicting training signal.

## 3.2 Retrieved and Augmented Image Encoding

A key component of our approach is encoding the $K$ conditioning images into sequences of conditioning tokens. We use a frozen CLIP ViT encoder Radford et al. (2021) to extract patch-level features $F_i = E(I_i)$ from each image $I_i$, providing rich, spatially descriptive representations beyond a global embedding. To condense this information efficiently, we apply a learnable Resampler $\Theta_R$, inspired by the Perceiver Resampler Jaegle et al. (2021) and IP-Adapter variants Ye et al. (2023). $\Theta_R$ maps $F_i$ to a compact set of $N_t = 16$ tokens, $T_i = \Theta_R(F_i)$, using a small set of learnable queries attending to $F_i$. These token sequences are then used to condition the diffusion model via cross-attention, balancing expressiveness with computational efficiency.

## 3.3 Retrieval-Conditioned Multiview Diffusion

The encoded tokens are then fed into a multiview diffusion model, which extends a 2D text-to-image U-Net architecture for multiview generation. Following MVDream Shi et al. (2023b), we incorporate camera pose embeddings for geometric guidance and modify the U-Net's self-attention layers. These layers are inflated to operate jointly over features from all generated views, forming a 3D-aware self-attention mechanism that promotes multiview consistency.

While MVDream relies solely on text-based cross-attention, we replace this mechanism with a decoupled cross-attention module that incorporates encoded tokens from both the text prompt and the retrieved images. Specifically, the tokens from the conditioning images are processed by a dedicated, trainable cross-attention branch, yielding retrieval-guided features denoted as $f_{\text{ret}}$.

For this cross-attention branch, we follow the design of IP-Adapter Ye et al. (2023), where the U-Net query features $Q_i$, generated via a shared query projection $\theta_Q$, attend separately to keys and values from the retrieved tokens $T_i$ and the text embedding. The retrieved tokens are processed through learnable projections $\theta_{K_{\text{ret}}}$ and $\theta_{V_{\text{ret}}}$ to produce $f_{\text{ret}}$, while the text embedding is processed through frozen projections $\theta_{K_{\text{txt}}}$ and $\theta_{V_{\text{txt}}}$, inherited from a pretrained diffusion model, yielding $f_{\text{txt}}$. We note that the shared query projection $\theta_Q$ is also frozen.

This results in a decoupled cross-attention mechanism. During training, we integrate the text and retrieval features as $f = \lambda f_{\text{txt}} + f_{\text{ret}}$, where $\lambda$ is a hyperparameter. Empirically, we find that small values of $\lambda$ ease the adaptation of the newly introduced retrieval branch. These text-conditioning

modules are deliberately kept frozen to preserve the base model's strong prior for in-domain concepts. The dynamic trade-off between this prior and the external retrieval signal for OOD concepts is then handled by our prior-guided attention mechanism during inference (see Sec. 3.5).

### 3.4 2D AND 3D TRAINING MODES

Our full architecture is trained jointly using the two data modes described below:

**3D Data Mode: Multiview Reconstruction.** When training with 3D samples, the model reconstructs a set of predicted views given their camera poses $\mathcal{C}$. The U-Net's self-attention layers operate across all $N$ view latents, enforcing cross-view consistency. Each target view is conditioned on its camera pose, and the text prompt $p$ provides global guidance via its features $f_{\text{txt}}$. The visual tokens aggregated from all $K$ augmented conditioning images $\mathcal{I}_{\text{ret}}$ are used to compute the retrieval attention features $f_{\text{ret}}$, jointly guiding the generation of all $N$ target views. A multiview reconstruction loss, $\mathcal{L}_{MV}(\theta, p, \mathcal{C}, \mathcal{I}_{\text{ret}}, \mathcal{I}_{\text{pred}})$, is applied across all target views. Critically, the conditioning views $\mathcal{I}_{\text{ret}}$ simulate in-the-wild retrieval scenarios where images may share geometry but have different textures or vice versa. By reconstructing a canonical object from these varied simulated retrievals, the model learns to disentangle and selectively utilize shared geometric and appearance features.

**2D Data Mode: Held-out View Prediction.** When training with 2D samples, the objective is to predict the single held-out image $I_{\text{target}}$ based on the text prompt $p$ and tokens from the $K$ conditioning retrieved images $\mathcal{I}_{\text{ret}}$. In this scenario, as only a single target view is generated, the U-Net's self-attention layers inherently function as standard 2D self-attention, operating within that single view's features. The text prompt $p$ and the tokens from the retrieved images provide conditioning via $f_{\text{txt}}$ and $f_{\text{ret}}$ respectively. Crucially, no explicit camera pose information is provided. This held-out strategy forces the model to infer 3D relationships from unstructured 2D data. This data may often depict shared geometry or textures from different perspectives, which are noisy or view-incomplete.

### 3.5 INFERENCE PROCESS

At inference time, given an input text prompt $p$, we first retrieve the top $K$ relevant 2D images $\mathcal{I}_{\text{ret}}$ from our diverse 2D database using the BM25-based text similarity approach.

To improve relevance, we compute prompt-caption similarity and discard images below a threshold, yielding $K' \leq K$ images. If no images pass the threshold, we disable retrieval-attention and fall back to the base model. These $K$ images are then encoded into visual tokens as detailed in Section 3.2. Our trained multiview diffusion model then generates $N$ consistent views conditioned on the text prompt $p$ and the set of retrieved tokens, utilizing specified camera poses for the target views.

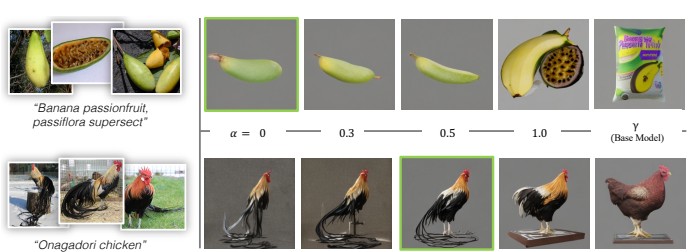

Figure 4: **Illustration of Prior-Guided Attention.** The base model's activations are leveraged proportionally to its prior knowledge of the object, controlled by the parameter $\alpha$. The results using prior-guided attention are marked with the green stroke.

**Prior-guided attention.** We introduce an adaptive fusion coefficient $\alpha$ that dynamically balances the influence of the model's prior knowledge and the retrieved signals, based on how OOD a prompt is. Diffusion models learn to approximate the score function $\nabla_x \log p(x|y)$, the gradient of the log data density which by definition points toward higher-probability regions of the data distribution Song & Ermon (2020). Thus, if a concept is in-domain, the base model's score will guide denoising toward an accurate reconstruction, whereas for OOD concepts the reconstruction will deviate.

To estimate $\alpha$ during inference, without ground-truth multiview, we first perform a short forward pass using only the base model's text-based attention $f_{\text{txt}}$ (retrieval module disabled) for 10 DDIM steps, generating an initial output. We then measure its similarity to retrieved images using DINOv2 similarity, which serves as a proxy for the base model's confidence in capturing the concept: high similarity indicates in-domain, so $\alpha$ favors $f_{\text{txt}}$; low similarity suggests OOD, shifting weight toward

Table 1: **Quantitative evaluation on OOD/rare concepts.** The models' performance is assessed on four orthogonal views. See Sec. 4 for further details.

| Method | 4-Views | | | | | Re-rendered (3D Reconstruction) | | | | |
|---|---|---|---|---|---|---|---|---|---|---|
| | CLIP ↑ | DINOv2 ↑ | IR ↑ | FID ↓ | IS ↑ | CLIP ↑ | DINOv2 ↑ | IR ↑ | FID ↓ | IS ↑ |
| **Text-to-3D** | | | | | | | | | | |
| MVDream | 66.47 | 33.12 | 58.01 | 76.71 | 10.62 | 70.83 | 28.66 | 58.98 | 96.29 | 11.39 |
| MV-Adapter (TX) | 66.48 | 28.53 | 58.42 | 84.28 | 9.55 | 71.33 | 24.30 | 56.14 | 106.66 | 11.23 |
| SPAD | 65.23 | 19.39 | 48.54 | 167.49 | 9.18 | 64.46 | 12.29 | 43.80 | 176.66 | 8.90 |
| TRELLIS (TX) | 67.96 | 21.11 | 51.01 | 160.93 | 6.90 | 67.16 | 16.87 | 51.82 | 154.43 | 8.09 |
| **Image-to-3D** | | | | | | | | | | |
| ImageDream-P | 69.20 | 45.01 | 65.64 | 68.40 | 12.11 | 70.44 | 32.77 | 60.17 | 103.24 | 12.84 |
| ImageDream-L | 67.55 | 39.48 | 63.93 | 84.69 | 9.45 | 70.16 | 29.60 | 58.66 | 120.37 | 10.42 |
| MV-Adapter (IM) | 69.74 | 49.14 | **71.05** | 72.71 | 12.88 | 71.53 | 35.25 | 60.36 | 107.95 | 12.64 |
| Era3D | 69.13 | 42.41 | 64.42 | 92.68 | **15.26** | 71.00 | 35.65 | 60.81 | 93.97 | **14.45** |
| TRELLIS (IM) | 70.31 | 35.24 | 59.32 | 167.61 | 11.38 | 67.86 | 24.43 | 52.23 | 146.82 | 10.72 |
| **3D Personalization** | | | | | | | | | | |
| MVDreamBooth | 66.14 | 36.22 | 55.09 | 82.73 | 11.55 | 68.38 | 27.91 | 54.33 | 107.07 | 11.92 |
| **MV-RAG (Ours)** | **71.77** | **50.19** | 67.41 | **54.79** | 13.20 | **74.28** | **39.61** | **66.59** | **80.54** | 12.33 |

the retrieval-based attention $f_{\text{ret}}$. The two sources are fused adaptively as: $f = \alpha \cdot f_{\text{txt}} + (\lambda' - \alpha) \cdot f_{\text{ret}}$, for a hyperparameter $\lambda'$, replacing the $f$ calculation used in training. The model is then run with the retrieval module enabled to generate final outputs. Fig. 4 illustrates its effect.

## 4 EXPERIMENTS

We evaluate our approach to state-of-the-art baselines on both OOD/rare and in-domain concepts.

**Benchmarks** As current benchmarks lack OOD/rare concept coverage, we curated 196 examples from Wikipedia Commons Wikimedia Commons (2025) (not used in training). Each example consists of a text prompt and multiple 2D retrieved images of the same concept. Importantly, texts were chosen to be far from any text (or concepts) seen during training. We call this evaluation set "OOD-Eval". See Appendix Sec.A.9 for additional details and examples. We also consider in-distribution objects, demonstrating that our success in OOD concepts is not compensated by worse in-domain results. To this end we consider a curated set of 50 in-domain objects from Objaverse-XL Deitke et al. (2023a). For retrieval, we consider 2D images from the LAION-400M dataset Schuhmann et al. (2021). For each text, we retrieve four 2D images. Unlike for OOD-Eval, we also have corresponding ground truth multiview images. We call this evaluation set "IND-Eval".

**Baselines** We compare MV-RAG against three categories of state-of-the-art methods. (1) *Text-to-multiview* generation: MVDream Shi et al. (2023b), MV-Adapter Huang et al. (2024) (text-conditioned), SPAD Kant et al. (2024), and TRELLIS Xiang et al. (2025) (text-conditioned). (2) *Image-to-multiview* generation applied to the retrieved views: ImageDream Wang & Shi (2023), MV-Adapter Huang et al. (2024) (image-conditioned), Era3D Li et al. (2024), and TRELLIS Xiang et al. (2025) (image-conditioned). (3) *3D personalization*: we adopt MVDream's optimization-based personalization approach Shi et al. (2023b); Ruiz et al. (2023), applied to all $k$ retrieved views. Unlike prior work, MV-RAG leverages multiple ($k = 4$ in our experiments) retrieved in-the-wild images that may differ significantly in pose, setting, and object identity. To the best of our knowledge, MV-RAG is the first framework to effectively use such diverse multi-image inputs for object multiview generation. To ensure a fair comparison, we adapt baselines accordingly: for image-to-multiview methods, we prompt each model separately with every retrieved condition image and report the best-scoring output across them. This setup provides baselines with access to the same retrieval set while respecting their single-image conditioning design. Further details are provided in Appendix Sec.A.6.3.

### 4.1 QUANTITATIVE EVALUATION

**Metrics** We assess generation quality with Inception Score (IS) Barratt & Sharma (2018) and FID Heusel et al. (2018) on the output poses. To assess alignment to the in-

put text, a natural choice would be to consider the CLIP Radford et al. (2021) similarity between the input text and the output multiview images produced by the model.

However, we found that CLIP (specifically in image-text similarity) is unable to score rare/OOD concepts well, often assigning a low score for such text-image pairs. See Appendix A.6.4 and Fig. 12 for further discussion and illustration. This is also demonstrated in Zhu et al.. As such, we evaluate image-image similarity between the generated views and held-out ground truth retrieved examples from our evaluation benchmark. Specifically, we compute the average similarity using CLIP and DINOv2 Oquab et al. (2023). Additionally, we employ an *Instance Retrieval (IR)* model Shao & Cui (2022) specifically trained to embed images of the same object instance close together in feature space, making it a more suitable choice for assessing entity-level visual alignment.

Table 2: **User study.** User study depicting (Q1-Realism), (Q2-Alignment) and (Q3-3D Consistency) reporting MOS (1-low, 5-high).

|  | Q1 ↑ | Q2 ↑ | Q3 ↑ |
|---|---|---|---|
| MVDream | 1.96 | 1.85 | 3.24 |
| ImageDream-P | 2.25 | 2.6 | 3.03 |
| MV-RAG (Ours) | **4.12** | **4.44** | **4.44** |

To evaluate 3D consistency, we adopt the procedure of Wang & Shi (2023), measuring how well a *3D reconstruction* model aligns with our generated views. Specifically, we use 4 generated views and train a *feed-forward 3D reconstruction* model Tang et al. (2024a) on these "training views". We then render and evaluate 18 novel views (Re-rendered in Tab. 1), whose fidelity and alignment with the training views are assessed using the metrics described above. Inconsistent multiview generations are expected to degrade reconstruction quality, leading to lower fidelity and alignment scores. Since geometric inconsistency is a primary

Table 3: **Quantitative evaluation on in-domain concepts.**

| Model | PSNR↑ | SSIM↑ | LPIPS↓ | CLIP↑ | SigLIP↑ |
|---|---|---|---|---|---|
| **Text-to-3D** | | | | | |
| MVDream | **16.95** | 0.717 | 0.363 | 64.25 | 34.81 |
| MV-Adapter (TX) | 15.37 | 0.632 | 0.459 | 59.62 | 30.18 |
| SPAD | 8.34 | 0.619 | 0.468 | 61.32 | 29.12 |
| TRELLIS (TX) | 16.53 | **0.743** | **0.327** | 60.67 | 30.98 |
| **Image-to-3D** | | | | | |
| ImageDream-P | 15.50 | 0.728 | 0.400 | 60.89 | 31.67 |
| ImageDream-L | 15.64 | 0.732 | 0.393 | 61.72 | 32.52 |
| MV-Adapter (IM) | 15.24 | 0.646 | 0.448 | 61.46 | 32.00 |
| Era3D | 12.44 | 0.722 | 0.378 | 58.79 | 29.94 |
| TRELLIS (IM) | 16.02 | 0.741 | 0.378 | 55.39 | 26.72 |
| **3D Personalization** | | | | | |
| MVDreamBooth | 16.31 | 0.716 | 0.381 | 61.68 | 32.14 |
| MV-RAG (Ours) | 16.63 | 0.730 | 0.362 | **64.48** | **35.34** |

failure mode for diffusion models on OOD concepts, this directly measures our method's ability to generate coherent 3D objects for rare prompts. Further details are in Appendix Sec.A.8.

**User study.** We complement our quantitative metrics with a user study on OOD-Eval prompts. Participants rated sets of four generated view from our model and baselines on a 1-5 scale for *Realism* (Q1), *Text Alignment* (Q2), and *3D Consistency* (Q3). See Appendix A.8 for full details.

**Evaluation on OOD/rare concepts** As shown in Tab.1, MV-RAG achieves strong performance across both evaluation modes. In the *4-views* setting, it outperforms all baselines on CLIP, DINO, and FID, while ranking second on IR (behind MV-Adapter (IM)) and IS (behind Era3D). In the more challenging *rerendered* setting which also reflects 3D consistency, MV-RAG leads on CLIP, DINO, IR, and FID, with Era3D attaining a higher IS. Notably, MVDream and ImageDream, which share similar architectures but lack retrieval, consistently underperform across metrics. The user study results in Tab.2 further corroborate MV-RAG's advantage,

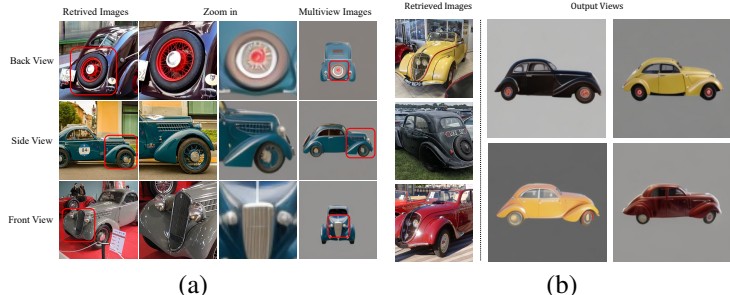

(a)             (b)

Figure 5: (a). **Utility.** On the LHS, we show retrieved views. The middle columns are zoom ins, for aspects used in generation, and the RHS shows back view (top), side view (middle), and front view (bottom). (b). **Diversity.** For the prompt "Peugeot 202", the LHS shows retrieved views, and the RHS shows a single view output (using the same pose) for 4 different seeds.

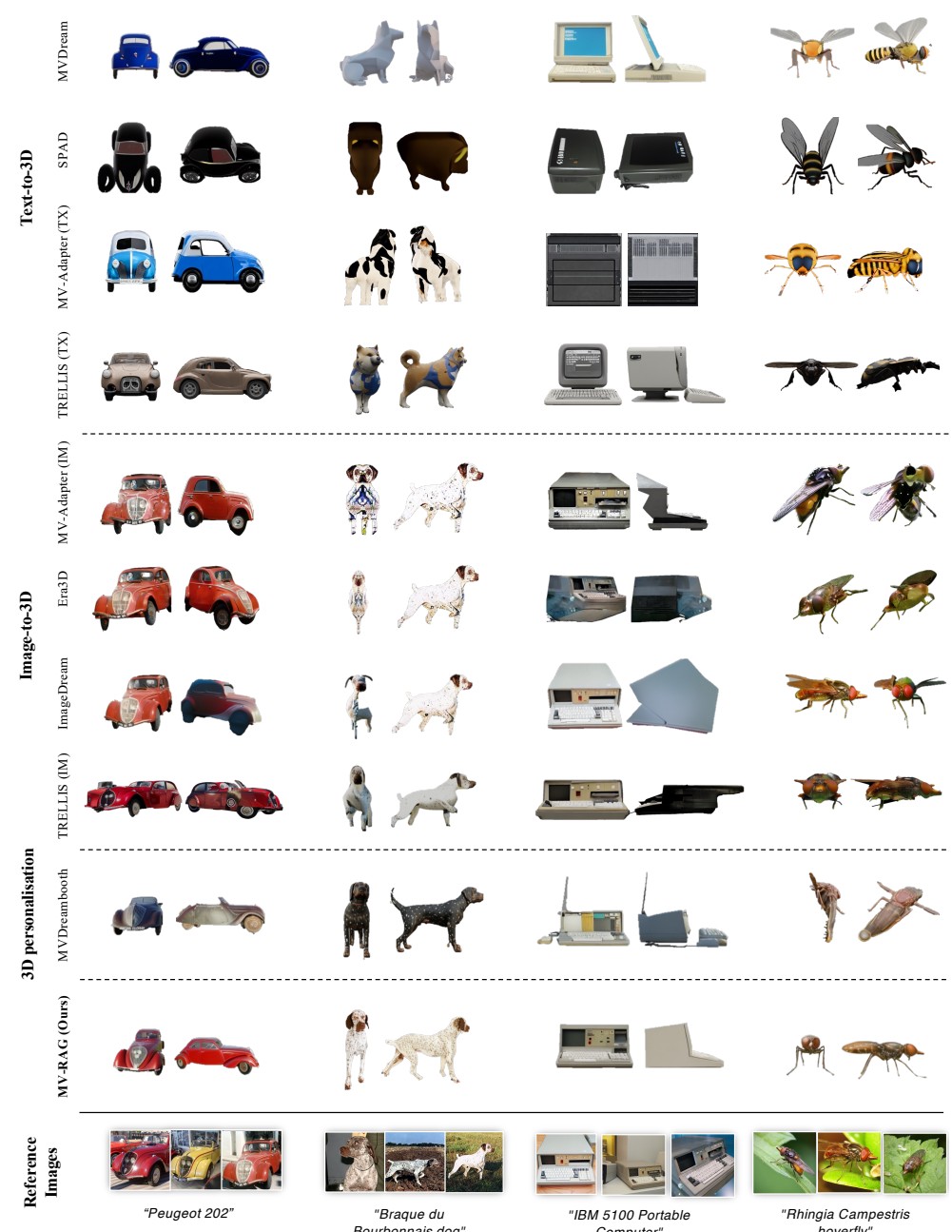

Figure 6: **Qualitative evaluation.** Text-to-3D models fail to generate unseen (OOD) concepts, while image-to-3D models fail to reconstruct their correct 3D structure from a single view. Existing personalization methods cannot effectively leverage the diversity of retrieved images.

showing clear gains in realism, text alignment, and 3D consistency. Our model's superior 3D consistency on OOD concepts, shown in the "Re-rendered" evaluation, is a direct result of our training. Unlike baselines whose internal priors are insufficient for OOD prompts, MV-RAG is explicitly taught in its 3D training mode to extract partial geometric cues from noisy, unposed views.

**Evaluation on in-domain concepts** We evaluate MV-RAG against all methods on the IND-Eval benchmark, which contains objects from the Objaverse Deitke et al. (2023a) dataset that is used for training in all baselines. Reconstruction quality is measured using PSNR, SSIM, and LPIPS with respect to the ground-truth views in IND-Eval, while text-image alignment is assessed via CLIP and SigLIP Zhai et al. (2023) similarity between the generated outputs and the input prompt. As shown in Tab. 3, MV-RAG achieves results that are on par with, or slightly surpass those of the baselines.

## 4.2 QUALITATIVE EVALUATION

Fig. 6 compares MV-RAG to baselines (see Appendix Figs. 18, 19 for more). Text-only models often fail on OOD objects, lacking visual priors and producing incorrect geometry. Single-reference image-to-3D methods are constrained by their single viewpoint: although they achieve high similarity to the input image, they cannot infer the true 3D structure for OOD objects, leading to smudging and artifacts in unobserved regions. Even multi-reference approaches like MVDreamBooth struggle to integrate diverse cues effectively, resulting in inconsistent colors, textures, and geometry. MV-RAG overcomes these limitations by leveraging multiple unposed images from a large 2D corpus, providing complementary viewpoints that enrich generation with relevant visual cues. Our framework isolates view-invariant attributes such as object texture while disentangling nuisance factors like illumination, occlusion, and background, producing diverse and accurate multiview outputs (Fig. 6).

**Diversity and Utility.** Unlike image-prompted methods, MV-RAG can produce diverse outputs for the same text by varying the random seed, as illustrated in Fig. 5(a). Moreover, Fig. 5(b) highlights MV-RAG's ability to leverage multiple retrieved views: the model combines information from different source views to generate consistent target views.

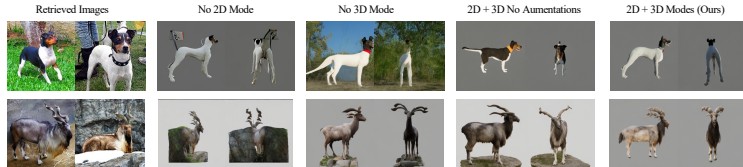

Figure 7: **Hybrid training ablations.** Output of *"Ratonero Bodeguero Andaluz dog"* (top) and *"Markhor goat"* (bottom) when our model is trained with (RHS) or without our 2D/3D schemes/augmentations.

## 4.3 ABLATION STUDIES

**Hybrid training** Fig. 7 presents a qualitative ablation of our 2D mode, 3D mode, and augmentations. Without the 2D mode, the model struggles to separate the object from its in-the-wild background, leading to artifacts (e.g., a floating leash on a dog, a goat merged with a rock). Without the 3D mode, it fails to consistently distribute visual features across views, producing inaccurate shapes (e.g., tail or horn) and background inconsistencies. Removing augmentations still allows in-the-wild settings through the 2D mode but reduces robustness to high variance in retrievals, yielding incorrect 3D structures.

**Number of Retrieved Images** Fig. 8 shows the effect of the number of retrieved images on alignment and fidelity. Using four views yields the best performance, as multiple exemplars provide complementary cues about geometry and texture, helping the model capture 3D structure under varying conditions. Beyond four views, gains saturate, suggesting redundancy rather than additional useful information.

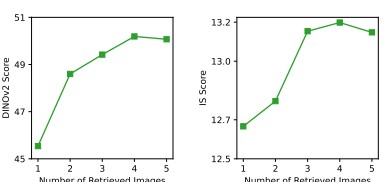

Figure 8: Effect of the number of retrieved images on alignment/fidelity.

**Additional Ablations** In Appendix A.7, we ablate the lexical *retrieval method* (BM25) and compare it against semantic dense retrievers (CLIP, SigLIP) on a combined corpus of OOD-Eval and MS-COCO. Appendix A.2 provides further ablations assessing (a). Retrieved images with *distinct appearances*, (b). *Noisy retrievals* (only $n < k$ relevant images), and (c). *Incorrect alpha scores* in attention weighting.

## 5 CONCLUSION

We introduce a retrieval-augmented multiview diffusion model for text-to-3D generation. By conditioning on relevant 2D images from a large database, our method produces consistent and accurate multiview outputs, particularly for out-of-domain (OOD) or rare concepts where prior methods struggle. A hybrid training scheme integrates structured multiview data with diverse 2D collections, using augmented conditioning views and a held-out view prediction objective. To evaluate challenging cases, we introduce a new OOD benchmark. Experiments show that our approach substantially improves 3D consistency, photorealism, and text alignment for OOD concepts while maintaining strong performance on standard benchmarks.

## 6 REPRODUCIBILITY STATEMENT.

To ensure reproducibility, we provide full implementation details in Appendix A.6, including network architectures, training procedures, and hyperparameters. The source code and the OOD-Eval benchmark are included in the supplementary material. Upon acceptance, all code, model weights, and the benchmark will be made publicly available, enabling researchers to reproduce our experiments and evaluate MV-RAG on both standard and out-of-domain scenarios.

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

Shuhan Zhu, Yonggang Zhang, Xinmei Tian, and Xiaoyan Sun. Prompt reverse learning: Enhancing visual language models for rare image recognition.

# A    APPENDIX

## A.1    ADDITIONAL QUALITATIVE EVALUATION

We provide further qualitative results to complement Fig. 6 from the main paper. Comparative examples are shown in Fig. 18 and Fig. 19, while additional outputs generated by our method are presented in Fig. 16 and Fig. 17.

## A.2    ADDITIONAL EXPERIMENTS

**Retrieved images with distinct appearances.**    We conducted a controlled experiment on OOD entities to test the effect of retrieval variance. We compared three strategies: (a) low variance (highly similar images), (b) moderate variance (our default top-k retrieval), and (c) high variance (deliberately diverse images). We observed that low variance yields accurate but less diverse generations, sometimes propagating instance-specific bias (see Sec. A.10 and Fig. 15)(a). Our default moderate variance offers the best balance of quality and diversity. High variance can sometimes challenge the model's ability to recover a consistent 3D structure, though the results remain superior to baselines given the same challenging inputs.

**Noisy retrievals (n < k relevant images).**    Our method is robust to retrieval noise. As described in Section 3.5, we employ a gating mechanism that filters out irrelevant images based on a similarity threshold. Specifically, for each retrieved image, if its BM25 score was lower than a given threshold (9.36), we do not consider it, and use the rest. Our performance in this case, therefore, translates to only using $K$ (=1,2,3, or 4) relevant retrieved images. This ablation is shown in Fig. 8, demonstrating that as $K$ increases, performance is improved.

**Incorrect alpha scores in prior-guided attention.**    The adaptive score $\alpha$ is critical. If $\alpha$ is underestimated for an OOD object, the model relies too heavily on its weak prior, and the output resembles a degraded baseline generation. Conversely, if $\alpha$ is overestimated, the model may overly trust the base model's prior even when it is flawed, potentially inheriting 3D structural errors (e.g., a floating tail), as discussed in Sec. A.10 and Fig. 15(c).

## A.3    CHALLENGES OF MULTIVIEW RAG IN COMPARISON TO SINGLE-VIEW RAG

While our approach builds upon the general idea of RAG, adapting it to multi-view generation introduces unique and substantial challenges that go beyond single-image RAG settings:

1. **Cross-view consistency from partial retrievals.** Single-image RAG only requires synthesizing a single image from the retrieved content. In contrast, multi-view RAG must ensure geometric consistency across several generated views. For example, if the model leverages a retrieved image that reveals the bonnet of a car for the front view, it must synthesize a side view that is consistent with that bonnet, even if the side view is not present or visible in any of the retrieved images. This kind of spatial reasoning is not required in single-view generation.

2. **View-specific and consistent use of different retrieved images.** Our method learns to draw on different retrieved images for different generated views, depending on which object components are visible. For instance, in Fig. 5(a), the model extracts the spare wheel, car grille, and front wheel from separate retrieved images and integrates them into different output views (back, front, and side). To do this effectively, the model must reason about 3D structure and spatial layout, a requirement absent from single-image RAG.

**How our method addresses (1) and (2)?** To address these issues, we designed a framework that can (i) selectively attend to relevant parts of the retrieved views for each target view and (ii) enforce consistency across generated images through shared conditioning and a hybrid training scheme. This is especially challenging because the retrieved views are in-the-wild and unposed, and no dataset provides direct supervision for multi-view consistency across retrieved parts.

Table 4: User study comparing the multiview outputs of MV-RAG against the CLAY (Rodin) demo. Our method was rated significantly higher in realism, alignment, and 3D consistency.

| Method | Q1: Realism ↑ | Q2: Alignment ↑ | Q3: 3D Consistency ↑ |
|---|---|---|---|
| CLAY (Rodin) | 2.20 | 2.40 | 3.95 |
| **MV-RAG (Ours)** | **4.60** | **4.60** | **4.63** |

## A.4 COMPARISON WITH CLAY (RODIN DEMO)

We also performed a qualitative comparison with CLAY Zhang et al. (2024), for which a public model was not available at the time of our evaluation. We used their publicly accessible online demo, Rodin[1], to generate multiview images for several prompts from our OOD-Eval set. We then conducted an additional user study following the protocol described above. Participants were shown randomly ordered sets of multiview images from Rodin and our method and were asked to rate them on a scale of 1-5 across three attributes. As summarized in Table 4, the outputs from MV-RAG were strongly preferred by users across all criteria. See Figure. 9 for comparison visualization.

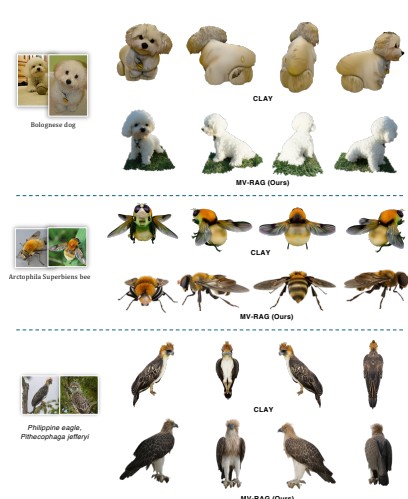

Figure 9: Qualitative Comparison against CLAY.

## A.5 LIMITATION OF SCALING MODELS

A natural question is whether out-of-distribution (OOD) challenges can be resolved simply by using a stronger model. To investigate this, we evaluate a strong text-to-image model (FLUX.1 Labs et al. (2025)) on several OOD prompts from our OOD-Eval benchmark (See Figure 10). Despite its scale and capacity, it frequently fails on these prompts, producing inaccurate structures, missing characteristic details, or hallucinating unrelated content.

This illustrates a fundamental property of OOD generalization. We argue that it cannot be solved solely by increasing model capacity or training data. OODness arises from distributional gaps rather than a fixed set of missing visual features understanding, and there will always exist concepts that fall outside a model's training distribution. Newly emergent or rare objects (e.g., a "Labubu doll") exemplify this challenge. Retrieval-based approaches such as MV-RAG address this limitation by conditioning generation on real visual exemplars, providing adaptability and robustness that scaling alone cannot achieve.

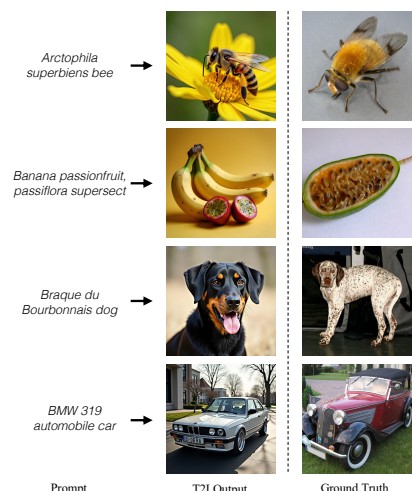

Figure 10: Large scale text-to-image model (FLUX.1) outputs on prompts from OOD-Eval.

---

[1] https://hyper3d.ai/

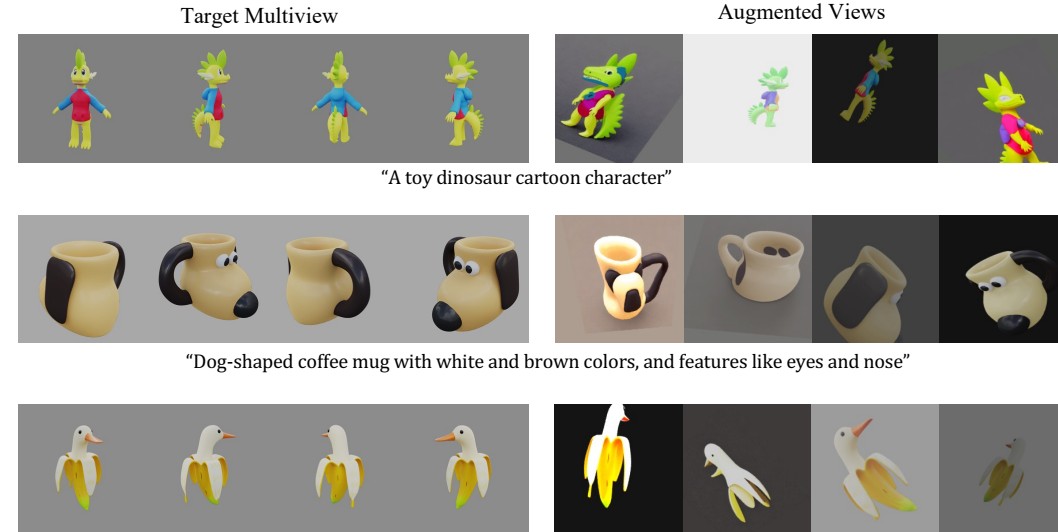

Figure 11: **Illustration of the augmented views in 3D mode.** Examples from the training set in 3D mode. For each target multiview instance, retrievals are simulated using photometric augmentations (crop, color jitter, perspective transform, etc.) and neural augmentations via an image-variation model.

## A.6 TRAINING, INFERENCE AND IMPLEMENTATION DETAILS

### A.6.1 DATA PREPARATION

For 3D mode training, we utilize multi-view synthetic renderings from the public Objaverse dataset Qiu et al. (2024); Deitke et al. (2023b) along with the associated camera parameters. We randomly sampled a subset of 90K objects. For each object, we select four orthogonal views with elevation angles in the range $[-5°, 30°]$ for supervision. Additionally, we sample 2-3 random views to simulate retrieval, as detailed below. Objaverse contains a wide variety of objects, including both high-fidelity, photorealistic assets and low-textured, abstract ones. To improve the model's robustness to real-world, non-synthetic data, we apply an aesthetic-based filtering criterion. This criterion incorporates color diversity, texture complexity, and multi-view consistency, which then results in about 65K objects. Following the preprocessing protocol of MVDream, we resize all rendered images to $256 \times 256$ pixels and replace empty backgrounds with a random gray color. Camera poses are normalized onto a unit sphere by removing translational components.

To simulate retrieval images, we apply a series of augmentations to the additional rendered views. These include perspective distortion, random rotations, resized cropping, and color jitter. To further enhance realism, we employ an image-variation model Ye et al. (2023); Rombach et al. (2022) to generate semantically and visually diverse variants of the same object. In total, we obtain four simulated retrieval images per object. We present an illustration for the augmentations in Fig. 11. For 2D mode training, we use the ImageNet21K datasetRidnik et al. (2021), which comprises over 21K semantic classes with multiple images per class. To improve the visual coherence within classes, we use a large language model (GPT-4o) to filter and retain only visually unified categories (e.g., *carpet shark*, *toilet bowl*) and exclude abstract or overly broad classes (e.g., *human*, *cycling*). This results in a curated subset of 516 visually consistent categories. For each selected class, we sample one target image for supervision and four additional images from the same class to serve as retrieved images.

### A.6.2 TRAINING

We fine-tune our model using the AdamW optimizerLoshchilov & Hutter (2019) with a learning rate of $5 \times 10^{-6}$ and a batch size of 24 for approximately 11,000 steps. Training is performed in an alternating scheme between 2D and 3D modes, allocating an equal number of steps to each mode. As

in MVDream, we append ", 3d asset" to the text prompt during 3D mode to help the model distinguish between the two training regimes. The model is initialized from the Stable Diffusion 2.1-based MVDream checkpoint, which remains frozen throughout training. The adapter modules are initialized from the ImageDream checkpoint. We fine-tune both the retrieval-attention modules and the Resampler. For the image encoder, we use OpenCLIP ViT-H/14, which is kept frozen during training. The training was done on a single NVIDIA A100 GPU, with a total training time of approximately 3 hours.

### A.6.3 BASELINES

For all baselines we use the official implementations and publicly available pretrained checkpoints provided by the respective authors, with the exception of MVDreamBooth, for which training code is not released. For each baseline, we generate 4 views using fixed orthogonal camera angles and elevations, employing the DDIM sampler with 50 steps and a classifier-free guidance (CFG) scale of 5. For image-to-3D baselines, we preprocess the retrieved reference images by segmenting out the background using Grounded-SAM Kirillov et al. (2023); Liu et al. (2023b); Ren et al. (2024). Among the reference images, we select the one that yields the highest multi-view consistency based on DINO score for evaluation. To ensure a fair comparison in image-image similarity metrics, we compare semantic features against the segmented ground-truth object views.

In figure. 14 we provide additional qualitative comparisons between MV-RAG and TRELLIS Xiang et al. (2025). For TRELLIS, we use the MV-RAG front-view output as its input. Even with a well-posed canonical view, the image-to-3D model fails to reconstruct the object when it is out-of-distribution, whereas MV-RAG leverages multiple in-the-wild retrieved images to handle such cases.

For the MVDreambooth baseline, we follow the method described in Shi et al. (2023b), training a separate MV-DreamBooth model for each instance in the OOD-Eval set. Each model is optimized for 600 steps. To preserve class identity, we apply a class-preservation loss using ImageNet class names (e.g., *dog*, *car*) when available, and default to the prompt when no corresponding class is derived.

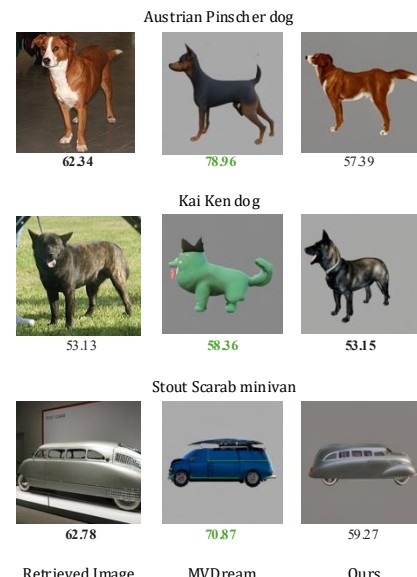

Figure 12: **Limitations of CLIP text-image similarity for evaluating OOD objects.** Each row shows an example from our OOD-Eval benchmark: the ground truth (GT) image, the output from MVDream, and the output from our model. Below each image is its CLIP similarity score. MVDream receives a higher score than both our model and the GT image, despite producing less faithful generations.

### A.6.4 METRICS

Figure 12 highlights three representative failure cases of using CLIP text-image similarity as an evaluation metric for out-of-distribution (OOD) objects. In each case, MVDream receives a higher CLIP score than both our model and even the ground truth image, despite generating outputs that are visually or semantically incorrect. We hypothesize that this stems from CLIP's limited prior knowledge of rare concepts and the fact that models like MVDream are optimized to align with CLIP-based features, potentially leading to overfitting to incorrect semantic associations. Further, as shown in Table 6, we find that for rare concepts, CLIP assigns nearly identical similarity scores to both detailed object names and their coarse class labels. This suggests that CLIP does not treat the additional semantic information in rare object names as meaningful, highlighting a lack of conceptual grounding for these OOD categories. In contrast, for in-domain objects, CLIP shows much stronger separation between specific and generic labels, reinforcing its limitations in recognizing and evaluating uncommon or unseen concepts.

These observations underscore the limitations of using CLIP for OOD evaluation and motivate our decision to adopt image-image similarity metrics instead, which more reliably reflect visual fidelity. To this end, we employ CLIP Radford et al. (2021), DINOv2 Oquab et al. (2023), and an Instance Retrieval (IR) model Shao & Cui (2022) fine-tuned from CLIP to better align visual object instances.

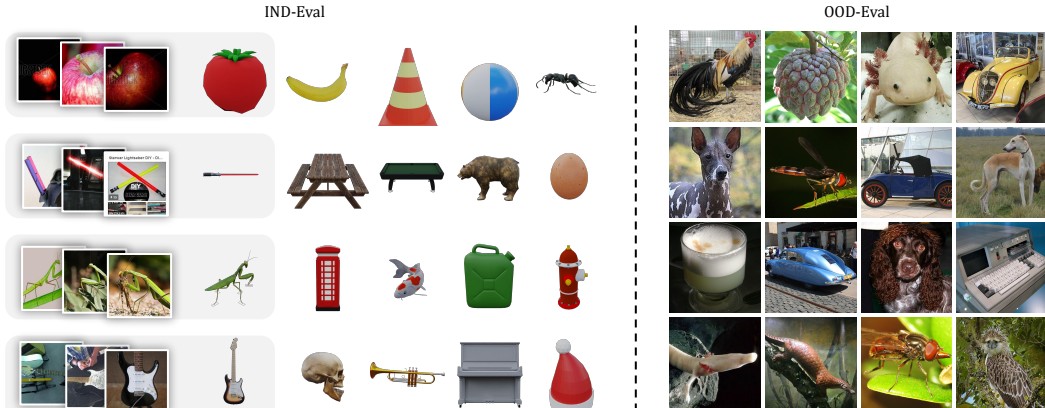

Figure 13: **Evaluation Benchmarks Overview. OOD-Eval:** Our out-of-distribution benchmark includes 2D images of both rare and well-known objects, featuring a diverse set of categories such as animals, vehicles, insects, foods, and everyday items. **IND-Eval:** The in-domain benchmark focuses on common, everyday objects that are representative of standard training distributions.

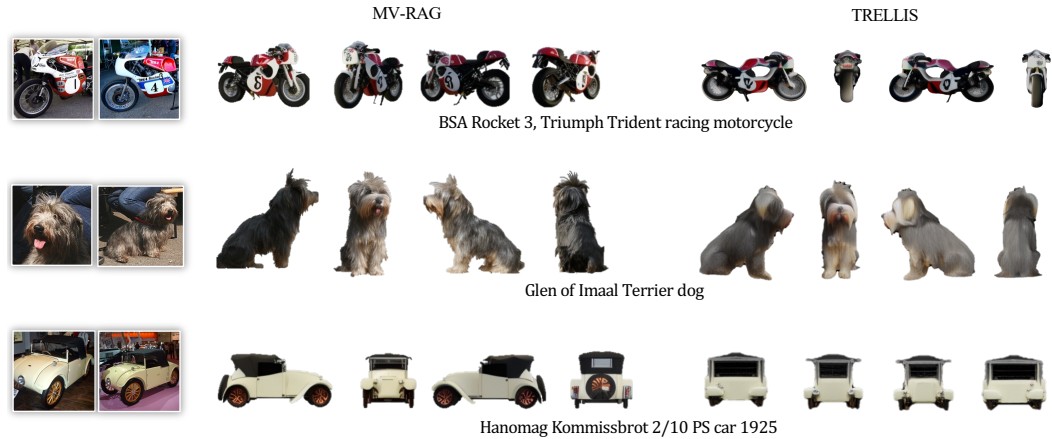

Figure 14: **Qualitative comparison against image-to-multiview with canonical-pose input image.** We present MV-RAG outputs against image-to-3D model over canonical pose image obtained from MV-RAG output

### A.6.5 RE-RENDERING

To more thoroughly assess the 3D consistency and fidelity of the baselines on the OOD-Eval benchmark, we employ LGM Tang et al. (2024a). Specifically, we reconstruct a 3D Gaussian representation using four output views generated by each model. From this reconstruction, we render 18 additional novel views sampled along a circular trajectory. These re-rendered views are then evaluated against the retrieved images using the image-image similarity metrics described earlier. We utilize the publicly available LGM implementation with its default configuration settings.

| Model | CLIP | DINO | IR | FID | IS |
|---|---|---|---|---|---|
| No 2D-mode | 72.995 | 39.100 | 66.196 | 82.0468 | **13.0656** |
| No 3D-mode | 73.435 | 37.973 | 65.557 | 84.6144 | 12.3207 |
| MV-RAG | **74.278** | **39.608** | **66.588** | **80.5406** | 12.3259 |

Table 5: Quantitative ablation study.

Table 6: **CLIP similarity between retrieved images and concept labels.** We compute similarity to the GT retrieved images with the object name against the class label (e.g., "dog", "car") and report average, max, and their absolute difference. OOD examples show minimal semantic separation.

| Domain | Text | Avg | Max |
|---|---|---|---|
| OOD | Bucovina Shepherd Dog | 63.59 | 67.16 |
|  | Dog | 63.41 | 67.45 |
|  | *Abs. Diff* | ***0.18*** | ***0.29*** |
|  | BMW 319 automobile car | 66.33 | 71.11 |
|  | Car | 65.14 | 71.56 |
|  | *Abs. Diff* | ***1.19*** | ***0.45*** |
| In-Domain | Airedale Terrier dog | 89.54 | 96.58 |
|  | Dog | 64.23 | 69.99 |
|  | *Abs. Diff* | *25.31* | *26.59* |
|  | American Hairless Terrier dog | 83.08 | 92.79 |
|  | Dog | 64.59 | 70.11 |
|  | *Abs. Diff* | *18.49* | *22.68* |

### A.6.6 TRAINING ABLATION STUDY

Table. 11 reports the re-rendered-view similarity scores for the three training configurations: no 2D mode, no 3D mode, and full MV-RAG. These metrics are computed on novel views produced by the reconstruction-and-re-rendering pipeline described above, and therefore capture both image-level fidelity and multi-view consistency.

### A.7 RETRIEVAL PROCESS AND ABLATION

We evaluate multiple retrieval strategies based on both image-text and text-text similarity.

For embedding-based retrieval with CLIP Radford et al. (2021) and SigLIP Zhai et al. (2023), we build an index

Table 7: Comparison of retrieval approaches for out-of-domain retrieval.

| Method | Precision@5 ↑ |
|---|---|
| CLIP (TX-IM) | 0.5366 |
| CLIP (TX-TX) | 0.7306 |
| SigLIP (TX-IM) | 0.7889 |
| BM25 (TX-TX) | **0.8522** |

using the FAISS library Douze et al. (2025), which supports efficient approximate nearest-neighbor search in high-dimensional spaces. We additionally employ Pyserini Trotman et al. (2014); Robertson et al. (1994); Lin et al. (2021) for text-based retrieval using the BM25 ranking function. This approach is a highly optimized and scalable toolkit designed for large-scale retrieval tasks, capable of indexing millions of documents while providing fast query responses. Its retrieval time is typically sub-linear with respect to corpus size due to inverted index structures, enabling near real-time search performance with minimal computational overhead, as demonstrated in large-scale search engine systems. We report the average inference time against all baselines in Table. 8.

**Ablation Study.** To assess the impact of the retrieval method on downstream generation, we compare BM25 with semantic dense retrievers: CLIP ( Radford et al. (2021) ) and SigLIP ( Zhai et al. (2023)) on a combined corpus of OOD-Eval and MS-COCO Lin et al. (2014). Table 7 reports the results. We evaluate retrieval performance using OOD-Eval text prompts, where the task is to retrieve the correct image-text pair from the combined collection. Specifically, we consider CLIP

with text-to-image (TX-IM) and text-to-text (TX-TX) similarity, SigLIP with TX-IM similarity, and BM25 as a lexical baseline.

We find that dense semantic retrievers often underperform in the OOD setting due to limited training exposure or weak grounding. As illustrated in Fig. 12 and Table 6, CLIP frequently assigns high similarity scores to incorrect or overly generic matches, failing to distinguish rare object names from broad categories. This weakness propagates to retrieval, where OOD queries often return irrelevant results. In contrast, BM25, which relies purely on lexical overlap, is more robust in these scenarios by directly matching rare keywords, without depending on learned semantic priors.

Table 8: Average inference time for each model.

| Model | Time (seconds) |
|---|---|
| MVDream | 1.081 |
| ImageDream | 1.470 |
| MV-Adapter | 6.562 |
| SPAD | 10.250 |
| Trellis | 12.098 |
| Era3D | 12.640 |
| MVDreambooth | 170.411 |
| MV-RAG | 6.296 |

### A.8 USER STUDY

We provide additional details about the user study referenced in Tab.2. The study involved 8 different objects, each evaluated using 3 methods: MV-RAG, MVDream, and ImageDream. For each object, participants were first shown a brief text description along with two sample images of the object to establish context. They were then shown sets of four images corresponding to different views-generated by each method. The internal order of the methods was randomized per object to mitigate ordering bias. Participants were asked to rate the following three questions on a scale from 1 to 5: (Q1) "How well do the 4 images match object?", (Q2) "How realistic do the 4 images look overall?", and (Q3) "How well do the 4 images appear to be consistent with each other, as if they show different views of the same 3D object?". The study was conducted using Google Forms, and participants viewed the images on a computer screen. The user population consisted of 30 randomly selected individuals across diverse ages, ethnicities, and genders.

### A.9 EVALUATION DATASET CONSTRUCTION

**Construction of OOD-Eval.** We construct an evaluation benchmark, OOD-Eval, consisting of 196 objects. To ensure diversity and out-of-distribution coverage, we use a large language model (GPT-4o) to curate object names representing rare or unique concepts, as well as familiar objects that are absent from the training data. These include examples such as extinct or rare animal species, uncommon vehicles, and other atypical items. A visual preview of the benchmark is provided in Fig. 13.

For generating image captions, we leverage a vision-language model, specifically Qwen-VL Bai et al. (2023) which provides high-quality textual descriptions of the images. These captions are used in the retrieval process (see Sec 3.5).

**Construction of IND-Eval.** We constructed an in-domain evaluation set by selecting 50 well-known or everyday objects from the widely used Objaverse-XL dataset. For each object, we retrieve 4 reference images from the large-scale LAION-400M dataset Schuhmann et al. (2021) using BM25-based text retrieval (see Sec. 3.5 in main paper). The retrieved images often exhibit significant visual or modality variation (e.g., artistic renderings or paintings of the object), as illustrated in Fig. 13.

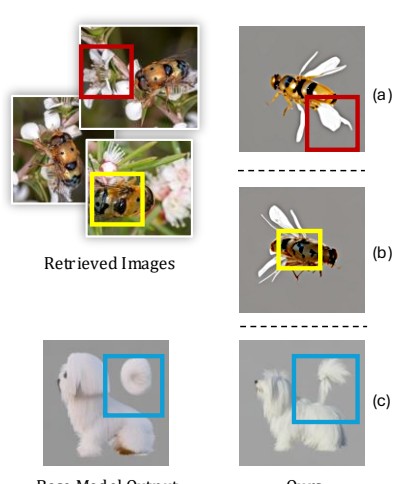

Figure 15: **Limitations.** (a) Visually biased retrieved images (e.g., repetitive white flowers) introduce artifacts in the generated multi-views. (b) The model struggles to reproduce fine-grained textures, such as the hoverfly's dorsal pattern. (c) When the base model (MV-Dream) is assigned a high attention weight ($\alpha$), 3D structural inaccuracies from the base model (e.g., a floating tail) are inherited.

Retrieved Images

Base Model Output · Ours

## A.10 LIMITATIONS

While effective, our method has several limitations. It relies heavily on both the quality of the retrieved image corpus and the capabilities of the underlying generative model, MVDream. When the base model lacks prior knowledge of the object and retrieval fails to provide informative or diverse references, the generated multiviews can be inaccurate or implausible.

As shown in Fig. 15, errors may arise when the retrieved images are visually biased-e.g., all showing similar white flowers, leading to reduced diversity and visual artifacts. Furthermore, our training objective promotes texture variation, which can make it difficult to reproduce fine-grained or specific patterns, such as the hoverfly's dorsal markings.

Our model also employs an adaptive mechanism that balances attention between the base model and the retrieval adapters, based on a similarity score between the generated initial views and retrieved images. When the base model demonstrates high similarity to the target object but exhibits 3D structural errors (such as a floating dog tail), these artifacts may be inherited. This limitation could be addressed by incorporating a more sophisticated and 3D-aware scoring function.

Further, while our current implementation generates four views, this is not a fundamental limitation of our retrieval-augmented framework, which could be applied to backbones that produce a larger set of views (e.g., Zero123++ Shi et al. (2023a)).

Lastly, our method introduces a retrieval phase prior to generation. Although this adds computational cost relative to standard text-to-image pipelines, the overhead is minimal. As shown in Section A.7

## A.11 ETHICS STATEMENT

Our work focuses on multiview image generation conditioned on text prompts and retrieved image corpora, with applications in graphics, virtual reality, and content creation. This technology can positively support immersive 3D visualizations and assist artists or designers in generating diverse object views from limited data.

At the same time, enhanced generative capabilities could be misused to produce highly realistic synthetic images, including disinformation or deepfakes. Biases present in the retrieval corpus or base models may propagate undesirable stereotypes or inaccuracies in generated outputs. To mitigate these risks, we recommend careful curation of datasets, transparency about generated content, and controlled access to highly capable models. We acknowledge and adhere to the ICLR Code of Ethics in the development and presentation of this work.

## A.12 LLM USAGE STATEMENT

A large language model (LLM) was used solely as a general-purpose writing assistant to aid in polishing and clarifying the text, including improving phrasing, grammar, and readability. The LLM was not involved in research ideation, technical development, or interpretation of results. All scientific content, analyses, and conclusions presented in this paper were independently verified and authored by the human authors, who take full responsibility for the accuracy and integrity of the work.

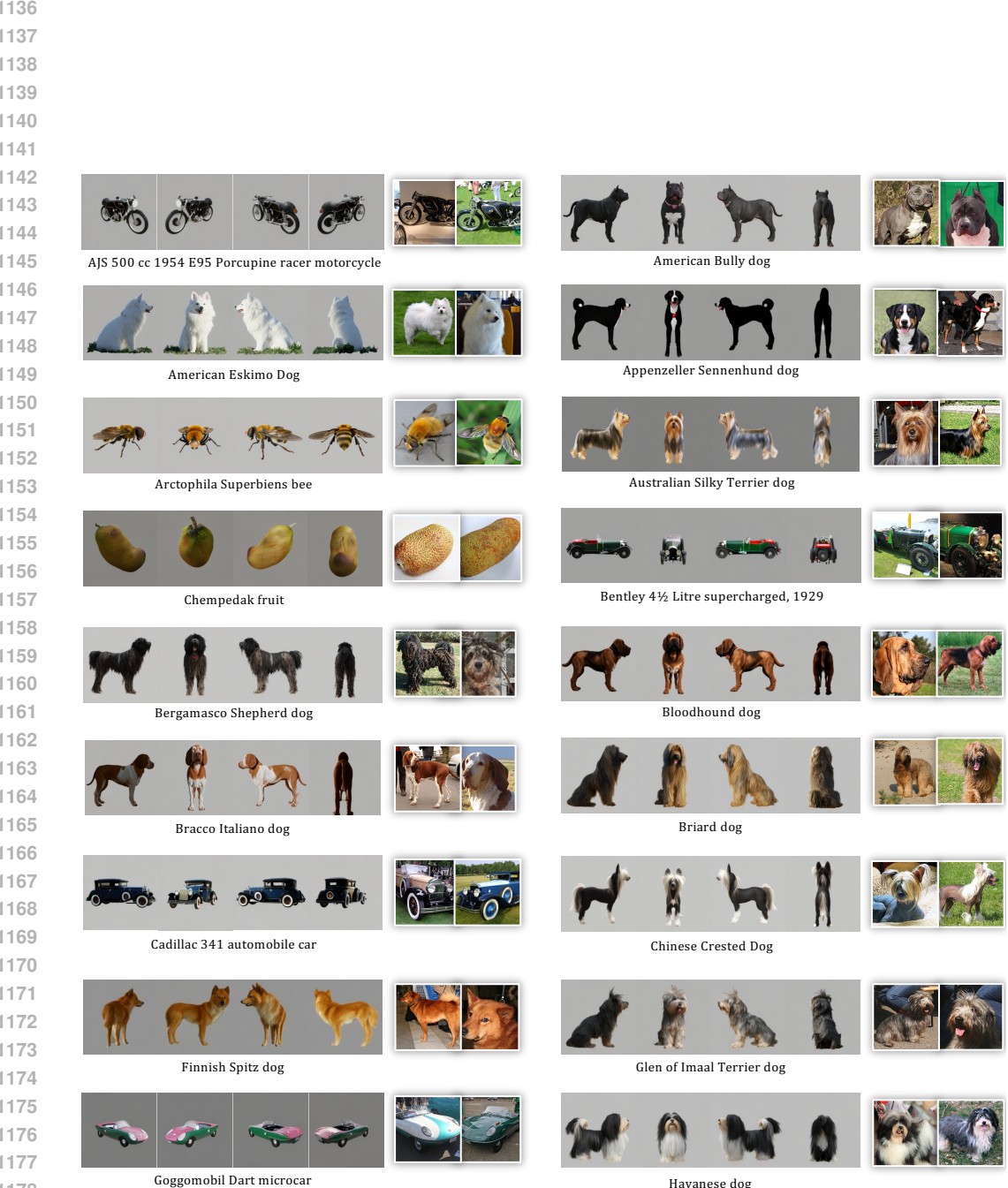

AJS 500 cc 1954 E95 Porcupine racer motorcycle

American Bully dog

American Eskimo Dog

Appenzeller Sennenhund dog

Arctophila Superbiens bee

Australian Silky Terrier dog

Chempedak fruit

Bentley 4½ Litre supercharged, 1929

Bergamasco Shepherd dog

Bloodhound dog

Bracco Italiano dog

Briard dog

Cadillac 341 automobile car

Chinese Crested Dog

Finnish Spitz dog

Glen of Imaal Terrier dog

Goggomobil Dart microcar

Havanese dog

Figure 16: **Additional Results.**

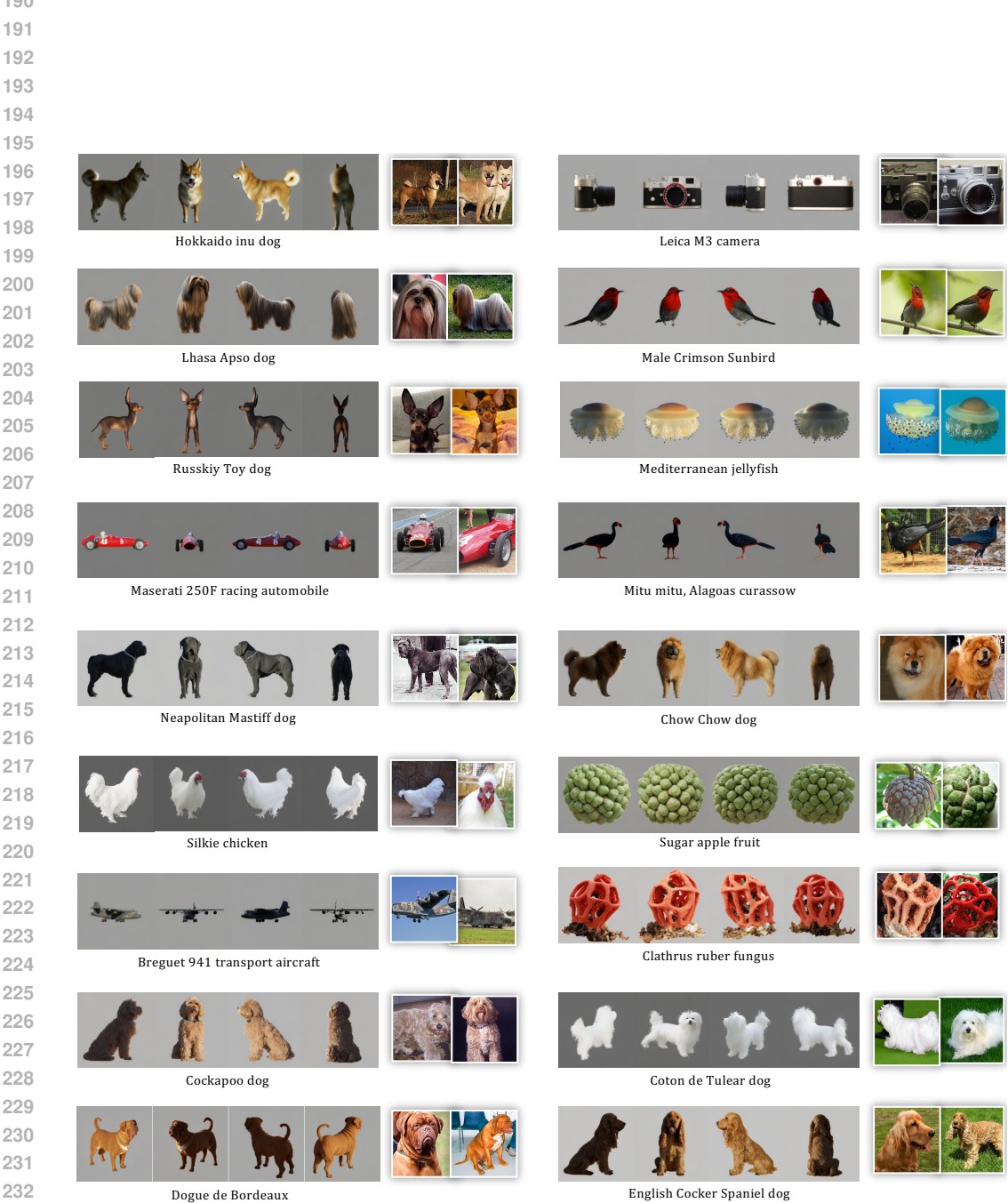

Figure 17: **Additional Results.**

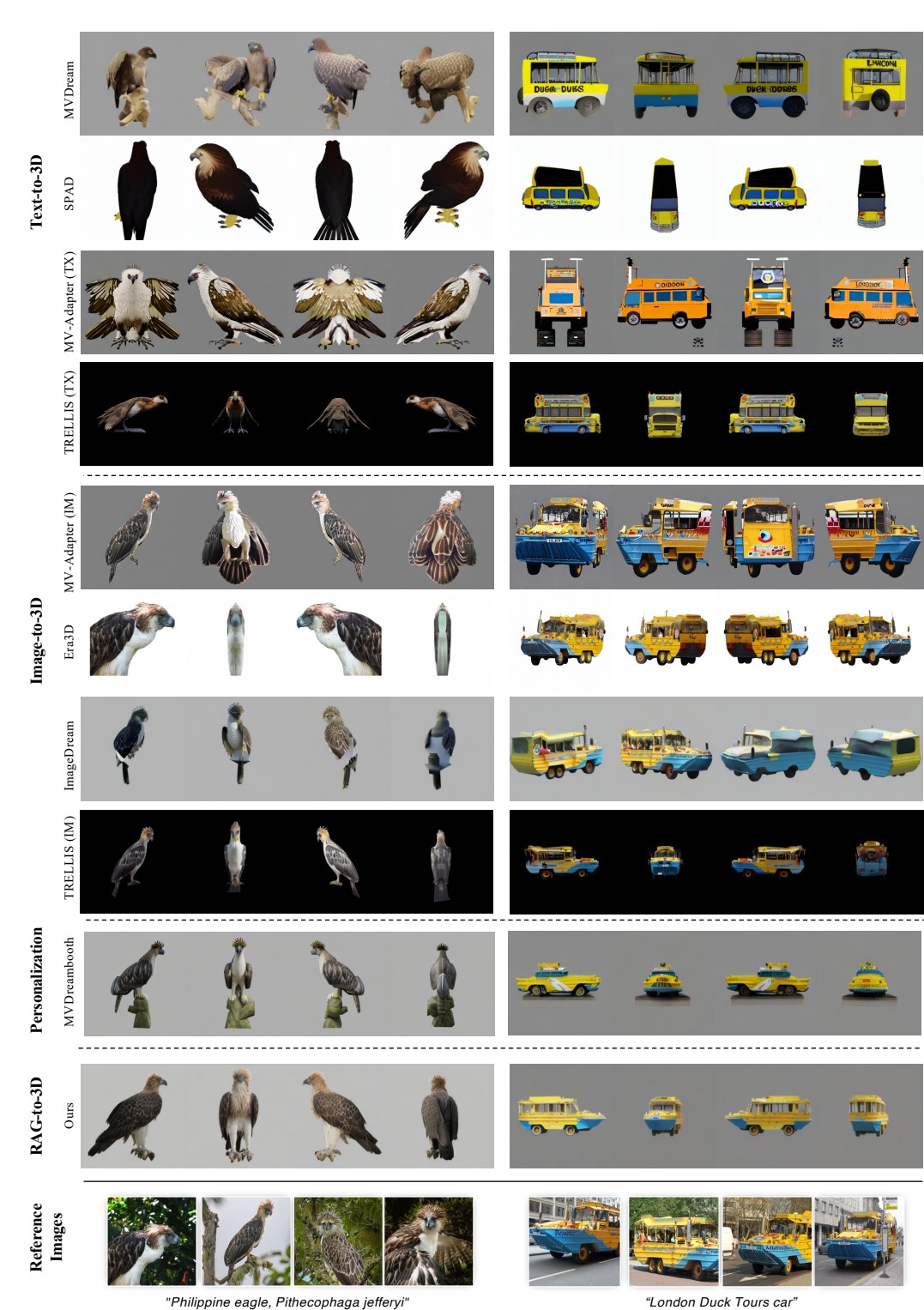

Figure 18: **Additional qualitative evaluation.** Additional examples to those shown in Fig. 6.

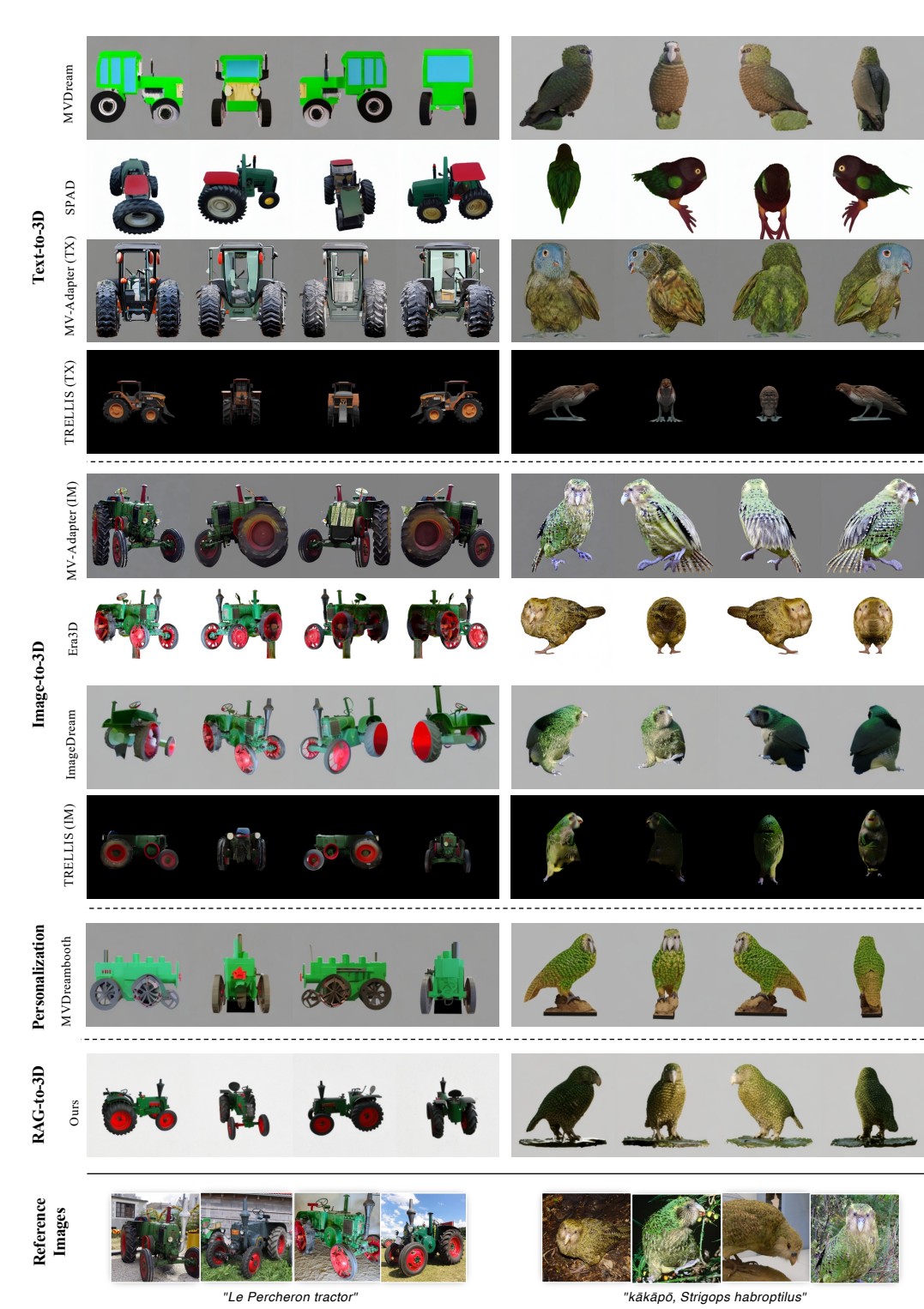

Figure 19: **Additional qualitative evaluation.** Additional examples to those shown in Fig. 6.

