# OpenReview forum: "MV-RAG: Retrieval Augmented Multiview Diffusion"
_ICLR.cc/2026/Conference — Submitted to ICLR 2026_

### Official Review · Reviewer_PLGn · 2025-10-30

**Soundness:** 3
**Presentation:** 3
**Contribution:** 3
**Rating:** 6
**Confidence:** 5

**Summary:**

This paper proposes MV-RAG, a retrieval-augmented multiview diffusion framework for text-to-3D generation, addressing the limitations of existing methods when handling out-of-distribution (OOD) or rare concepts. MV-RAG first retrieves relevant 2D images from a large in-the-wild database based on text prompts, then uses a hybrid training scheme (combining 2D held-out view prediction and 3D multiview reconstruction) to train the diffusion model, with an adaptive fusion coefficient balancing the base model’s prior and retrieval signals. The authors also introduce the OOD-Eval benchmark for rigorous OOD evaluation. Experiments show MV-RAG improves 3D consistency, photorealism, and text alignment for OOD/rare concepts while maintaining competitive performance on in-domain benchmarks.

**Strengths:**

1. The introduction of the RAG for multiview diffusion models is interesting, which adds some external information for the diffusion model.
2. The proposed method shows some promising performance on OOD examples.

**Weaknesses:**

1. Motivation is not so clear. If the retrieved images are also OOD for the multiview diffusion model, then the method also cannot accurately generate the 3D shape. If the retrieved images fall in the training domain, then an alternative way is just to generate one image using the input texts, and then an image-to-3D model can handle this task. So theoretically, this paper does not well resolve the generalization problem.
2. The comparison is not so fair. For baseline methods like TRELLIS, they may be sensitive to the viewpoints of the input images (in the image-to-3D setting). The retrieved input images show diverse perspective distortions and uncommon viewpoints with incomplete foreground objects. I guess this is the main cause for the bad performance of TRELLIS on the retrieved images. I think this is also a good research problem to address the viewpoint problem, but this is deviated from the original claim of the paper.

**Questions:**

Refer to weakness.

---

> ### Author Response · Authors · 2025-11-23
>
> **[part 1/2]**
>
> We sincerely thank the reviewer for the overly positive evaluation of our work. We are grateful for the recognition that introducing retrieval-augmented generation to multiview diffusion models is interesting and that our method demonstrates promising performance on out-of-domain examples. Below, we address the reviewer’s weaknesses and questions:
>
> **Regarding Motivation (Weakness 1):**
> We respectfully clarify our motivation with regard to the generalization problem. Our core argument is that the generalization gap cannot be addressed through additional finite training data alone. To demonstrate this, we conducted experiments with the large-scale state-of-the-art text-to-image model FLUX.1 on our OOD-Eval dataset (Appendix A.5, Figure 10), which clearly shows a knowledge gap for out-of-domain concepts even in such large-scale models.
>
> MV-RAG addresses this fundamental limitation through a RAG-based solution: rather than attempting to encode all possible visual knowledge during training (which is infeasible) using a parametric memory, we train the model to extract and leverage visual cues from the external retrieved images (non-parametric memory). This is a general, domain-agnostic methodology that can adapt to new concepts at inference time.
>
> **Reframing the OOD Problem:** The reviewer suggests that if retrieved images are OOD for the multiview model, generation will fail. We argue that this perspective applies only to purely parametric models, where the network weights must implicitly "store" every possible concept.
>
> * **Parametric Limitations:** A model's parametric memory is finite and static; it cannot store the infinite long tail of visual concepts. If a specific OOD concept is not in the weights, a standard model fails to generate it accurately.
>
> * **Non-Parametric Solution:** MV-RAG offloads this storage requirement to the non-parametric retrieval database, which can store the vast long tail of data. Our model is not trained to "memorize" these specific OOD examples; instead, it is trained to learn the **mechanism of geometric transfer**. By simulating retrieval variance during training (via augmentations in 3D mode and diverse sampling in 2D mode ), the parametric model learns how to interpret and "lift" arbitrary external visual cues into 3D. Thus, even if the semantic concept is completely unknown to the model's weights (OOD), the model succeeds by relying on the external knowledge provided by the retrieval.
>
>
> **Comparison to text-to-image-to-multiview pipeline:** To directly address the reviewer's alternative approach, we conducted an experiment using FLUX.1 (state-of-the-art text-to-image) followed by MV-Adapter (image-to-multiview). Results demonstrate that this pipeline significantly underperforms MV-RAG:
>
> | Method                          | CLIP ↑  | DINO ↑  | IR ↑     | FID ↓    | IS ↑   |
> |---------------------------------|---------|---------|----------|----------|--------|
> | Text-to-image-to-multiview     | 68.460  | 31.718  | 56.469   | 124.662  | 5.698  |
> | Ours (MV-RAG)                  | **71.770**  | **50.190**  | **67.410**   | **54.790**  | **13.200** |
>
> This validates that retrieval-augmented generation is essential for OOD concepts, as even state-of-the-art text-to-image models lack sufficient knowledge to generate these concepts accurately.

---

> > ### Author Response · Authors · 2025-11-23
> >
> > **[part 2/2]**
> >
> > **Regarding Fairness of Comparison (Weakness 2):**
> > We appreciate the reviewer raising this concern and would like to provide several clarifications:
> >
> > **Our evaluation protocol already accounts for viewpoint sensitivity:** For all image-input baselines, including TRELLIS, we retrieve 4 images and evaluate each separately, reporting only the **best-performing result** (Appendix A.6.3). This gives baselines the advantage of selecting the most favorable viewpoint, making our comparison conservative rather than unfair.
> >
> > **TRELLIS with canonical viewpoints:** To further address the viewpoint concern, we conducted an additional experiment where we provided TRELLIS with canonical front-view images from MV-RAG's own outputs—viewpoints that align with those shown in the TRELLIS paper. Note that this comparison is actually **unfair to us** (TRELLIS receives idealized input while MV-RAG must handle diverse retrievals), yet TRELLIS still significantly underperforms:
> >
> > | Model                           | CLIP    | DINO    | IR      | FID       | IS       |
> > |---------------------------------|---------|---------|---------|-----------|----------|
> > | TRELLIS (Using MV-RAG output)   | 70.400  | 22.052  | 52.251  | 195.9709  | 8.6747   |
> > | MV-RAG                          | **74.583** | **46.809** | **67.180** | **53.9550** | **13.5840** |
> >
> > Qualitative examples are provided in Appendix Figure 14.
> >
> > For reference, we also evaluated TRELLIS using the same experimental setting (canonical views from MV-RAG output) on **in-domain objects**, where TRELLIS performs comparably to MV-RAG:
> >
> > | Model                           | CLIP     | DINO     | IR       | FID        | IS        |
> > |---------------------------------|----------|----------|----------|------------|-----------|
> > | TRELLIS (Using MV-RAG output)   | 73.823   | 39.064   | 54.445   | 206.5819   | **6.3562** |
> > | MV-RAG                          | **74.891** | **52.800** | **66.516** | **132.1390** | 5.5155    |
> >
> > These results demonstrate that the performance gap is **not primarily due to viewpoint sensitivity**, but rather reflects fundamental limitations in handling out-of-domain concepts—which is precisely the problem MV-RAG is designed to address.
> >
> > **Multi-image input TRELLIS:** We also tested TRELLIS with multiple image inputs, but performance remained inferior to MV-RAG. Since TRELLIS was not specifically trained to handle multi-image input, nor presented in the TRELLIS paper, we omitted this comparison from the paper.
> >
> > We thank Reviewer UQp6 for the constructive feedback and clear understanding of MV-RAG’s contributions—especially the novelty of retrieval-augmented multiview diffusion and our hybrid 2D–3D training scheme. The questions on retrieval sensitivity and model scaling helped us further strengthen the paper. We believe our responses address the concerns raised and are happy to provide any further clarification during the rebuttal period.

---

### Official Review · Reviewer_UQp6 · 2025-10-31

**Soundness:** 3
**Presentation:** 3
**Contribution:** 3
**Rating:** 6
**Confidence:** 2

**Summary:**

This paper presents MV-RAG, a retrieval-augmented multiview diffusion framework for text-to-3D generation. The method retrieves relevant in-the-wild 2D images based on the input text and uses them to condition a multiview diffusion model, producing geometrically consistent and photorealistic multi-view outputs. A hybrid 2D–3D training strategy combines structured multiview supervision with unposed 2D data through a held-out-view prediction objective.

**Strengths:**

The application of retrieval-augmented generation to 2D–3D diffusion is novel and well-motivated.
    The proposed hybrid 2D–3D training scheme is technically sound and effectively integrates structured and unstructured supervision.
    The paper is thorough and well-organized, featuring extensive ablation studies, user evaluations, and clear architectural details.

**Weaknesses:**

Although qualitative results are strong, the quantitative improvements on standard metrics (e.g., PSNR, IS) are modest—noticeable but not substantial.
    The evaluation could benefit from comparisons with more diverse and standardized baselines such as Wonder3D or SyncDreamer.

**Questions:**

How sensitive is MV-RAG’s performance to retrieval quality or the domain coverage of the retrieval database?
    The paper mentions fine-tuning on a single GPU for three hours using a Stable Diffusion checkpoint. Were larger-scale experiments conducted? In particular, would increasing the dataset size or model capacity further improve performance?

---

> ### Author Response · Authors · 2025-11-21
>
> We sincerely thank the reviewer for the overly positive evaluation of our work. We are grateful for recognizing that our application of retrieval-augmented generation to multi-view diffusion is novel and well-motivated, and that our hybrid 2D-3D training scheme is technically sound. We appreciate the acknowledgment of our paper's thoroughness and comprehensive experimental validation. Below, we address the reviewer’s weaknesses and questions:
>
> **Quantitative improvements**: We would like to clarify the context of our quantitative evaluation. PSNR is measured on in-domain objects from Objaverse, which represents the same distribution as our baselines' training data. In this setting, we demonstrate comparable or even slightly improved performance to existing methods. However, our key contribution lies in achieving state-of-the-art results on out-of-domain (OOD) concepts, where our model significantly outperforms baselines. We refer the reviewer to lines 427-431 for a detailed discussion of this evaluation design.
>
> Regarding Inception Score (IS), we note that IS captures both image quality and diversity. Since our evaluation focuses on out-of-domain objects generating 3D outputs with uniform backgrounds, diversity is not our primary concern. We include FID as our main quality metric (which measures perceptual quality without the diversity component) and retain IS for completeness and comparison with prior work.
>
> **Additional baselines**: We appreciate this suggestion. Our experimental comparison includes recent state-of-the-art methods (TRELLIS, CLAY, MV-Adapter) as well as architecturally similar approaches (MVDream, ImageDream, MVDreambooth, MV-Adapter). While Wonder3D and SyncDreamer are relevant works, they have been quantitatively compared against our selected baselines in subsequent publications (e.g., Era3D, ImageDream), where our baselines demonstrated superior performance. We therefore believe our baseline selection captures the current state-of-the-art effectively.
>
> **Retrieval quality sensitivity**: We address this important question in Appendix Section A.2. MV-RAG incorporates a gating mechanism that filters retrieval images based on their relevance scores. When retrieved images are noisy or irrelevant, they are automatically discarded. Consequently, if no relevant images exist in the retrieval corpus, MV-RAG falls back to the base text-to-multiview model, relying solely on the text prompt. This design ensures robust performance across varying retrieval quality.
>
> **Larger-scale experiments**: We agree that additional training data and larger model capacity would likely benefit performance. We conducted experiments with both SD1.5 and SD2.1 as base models, and observed that upgrading from SD1.5 to SD2.1 indeed improved results. We emphasize that MV-RAG's architecture introduces lightweight adapters to existing pretrained models rather than training from scratch. This design choice is deliberate: it allows us to leverage the extensive knowledge already encoded in large-scale pretrained diffusion models while not requiring large-scale training data for the retrieval modules. Moreover, because our method relies on lightweight plug-and-play adapters, MV-RAG can flexibly incorporate any stronger or more recent diffusion backbone, making improvements in the underlying model capacity immediately transferable to our system.
>
> We thank Reviewer UQp6 for the constructive feedback and clear understanding of MV-RAG’s contributions, especially the novelty of retrieval-augmented multiview diffusion and our hybrid 2D–3D training scheme. The questions on retrieval sensitivity and model scaling helped us improve the paper with additional clarifications and experiments. We believe our responses address the concerns raised and remain available for further clarification during the rebuttal period.

---

### Official Review · Reviewer_fdtJ · 2025-10-31

**Soundness:** 3
**Presentation:** 3
**Contribution:** 3
**Rating:** 4
**Confidence:** 2

**Summary:**

MV-RAG introduces a retrieval-augmented multi-view diffusion framework for text-to-3D generation, specifically addressing out-of-distribution (OOD) concepts. The approach conditions generation on relevant in-the-wild 2D images retrieved for each text prompt, employing a hybrid training scheme that combines 3D geometric consistency objectives with 2D held-out view prediction. During inference, a prior-guided attention mechanism dynamically balances the base model's internal knowledge with external retrieval signals. Experiments demonstrate MV-RAG's superior performance over SOTA baselines in 3D consistency, photorealism, and text alignment on OOD benchmarks while maintaining competitive in-domain results.

**Strengths:**

Quality & Originality: The primary strength of this paper lies in its exceptionally comprehensive and multi-faceted experimental evaluation, which leaves little room for doubt regarding the efficacy of the proposed MV-RAG framework. The experimental section is thorough and systematic, and the experiments are designed to systematically validate the method's performance across a wide range of scenarios and against numerous strong baselines.

(1)	The paper includes detailed ablation experiments on a hybrid 2D/3D training scheme, the number of retrieved images(K), the retrieval method (BM 25 v.s. dense retrievers), noisy retrievals, and the importance of the prior-guided attention mechanism.

(2)	The author thoughtfully addressed the limitation of CLIP for evaluating OOD concepts and introduced a novel evaluation protocol (image-image similarity metrics as complementary).

(3)	The authors provided an exhaustive benchmarking against SOTA, including three different paradigms.

(4)	The author also provided an analysis of limitations and failure cases, honestly discussing the limitations and failure cases of their methods.

(5)	To complement quantitative metrics, the paper also includes a user study that directly assesses the critical attributes of realism, text alignment and 3D consistency from human perspectives.

Clarity: Beyond the technical contributions, the paper is well-written and structured. The clarity of exposition is supported by comprehensive appendices and supplementary materials. The complete OOD-Eval benchmark and code provided in the Supplementary Materials underscore a strong dedication to reproducibility.

(1)	Significance: The work has a significant impact, as evidenced by its SOTA performance on standardized benchmarks.

**Weaknesses:**

(1)	Ablating the distinct roles of 2D mode and 3D mode (Section 4.3)
The qualitative ablation in Figure 7 effectively illustrates the distinct roles of the 2D and 3D training modes. The accompanying text (lines 458-461) states that the 2D mode is crucial for separating objects from in-the-wild backgrounds, while the 3D mode is vital for correct shape rendering and background consistency.
However, the described effects that are specifically concerning background separation and shape correctness can sometimes be interdependent and challenging to distinguish purely from visual examples. To provide a more precise and quantifiable understanding for the reader, could the authors supplement the qualitative ablation with quantitative metrics designed to measure these specific aspects?

(2)	I appreciate the detailed description of the OOD-Eval benchmark construction in Appendix A.9 and the inclusion of the full dataset. The use of an LLM (GPT-4o) to curate "rare or unique" concepts is a practical approach.
However, I have a concern regarding potential category bias. The visualization in Figure 12 and a perusal of the supplementary data suggest a possible over-representation of certain categories, such as dogs and vehicles. I wonder if the LLM-based generation might inherently favor such semantically dense and well-documented categories.
This could potentially lead to an overestimation of the method's average performance on the true long tail, as these categories often have abundant, high-quality, multi-view images available for retrieval (which may make the retrieval task easier).

(3)	Computational Efficiency and Inference Latency
The paper rightly focuses on generation quality. However, for a complete understanding of the method's practical utility, its computational overhead needs to be addressed. The training efficiency is noted (∼3 hours on a single A100, line 848), which is commendable. Nevertheless, computational efficiency is mentioned briefly; a detailed comparison of end-to-end inference latency against baselines would help assess the method's practical utility.

**Questions:**

(1)	According to Weakness – (1), I kindly suggest the authors to provide the following experiment to quantify the effectiveness for 2D mode and 3D mode, which would be clearer to delineate the unique and complementary functionalities of each training mode.
a)	For object-background separation, a segmentation metric could be used to calculate the mIoU between generated objects and their backgrounds, assessing how well the model isolates the foreground.
b)	For shape correctness, the 3D consistency metrics already employed (e.g., the proposed re-rendered DINOv2 similarity) are a good proxy. A comparative analysis of this metric for models trained with and without the 3D mode could quantitatively underscore its contribution to geometric accuracy.

(2)	Concerning weakness - (2), to better characterize the benchmark and the method's performance across it, can the authors provide the following analyses:
a)	Category Distribution Analysis: Provide a statistical breakdown of the OOD-Eval concepts across broader semantic categories (e.g., animals, vehicles, furniture, artifacts, food). This would make the benchmark's composition transparent.
b)	Quantification of OOD: The paper states the texts were "far from any text... seen during training" (line 300). To substantiate this claim, could the authors quantify this distance? For instance, they could plot the distribution of CLIP text embedding distances between all OOD-Eval prompts and the training set prompts (or a large sample from LAION). Showing this distribution, and further segmenting it by the semantic categories from the first analysis, would provide a rigorous, quantitative measure of how "out-of-distribution" the benchmark truly is.

(3)	According to weakness – (3), can the authors provide the following results:
a)	A table reporting the average time from input text prompt to the output of 4 generated views for MV-RAG and its key competitors. This would allow readers to better assess the trade-off between the achieved quality gains and the associated computational cost.

---

> ### Author Response · Authors · 2025-11-23
>
> **[part 1/3]**
>
> We thank the reviewer for the thorough evaluation and recognition of the quality and originality of our work. We appreciate the acknowledgment of our comprehensive experiments—including ablations, the OOD evaluation protocol, and extensive benchmarking—as well as the clarity, reproducibility efforts, and discussion of limitations. Below, we address the reviewer’s remaining questions and concerns:
>
> **Addressing point (b) - Shape correctness for 3D mode:** We thank the reviewer for suggesting this evaluation and fully agree that the 3D consistency metrics effectively quantify this contribution. We provide the requested re-rendered evaluation in the table below (added as Appendix Table 5). As can be seen, the model trained without 3D mode demonstrates degraded performance across multiple metrics on re-rendered outputs, particularly in DINOv2 similarity (37.973 vs. 39.608 for full MV-RAG), confirming the 3D mode's critical role in geometric accuracy and multi-view consistency.
>
> | Model      | CLIP     | DINO     | IR       | FID       | IS        |
> |------------|----------|----------|----------|-----------|-----------|
> | No 2D-mode | 72.995   | 39.100   | 66.196   | 82.0468   | **13.0656** |
> | No 3D-mode | 73.435   | 37.973   | 65.557   | 84.6144   | 12.3207   |
> | MV-RAG     | **74.278** | **39.608** | **66.588** | **80.5406** | 12.3259   |
>
> **Addressing point (a) - Object-background separation for 2D mode:** We appreciate the suggestion of mIoU; However, we found that segmentation metrics are unreliable for generated OOD content because we lack pixel-aligned ground truth masks for generated samples. Therefore, scale variations would confound measurements, e.g., given a “GT” image of an object, a smaller generated clean object image vs. a larger generated object with background artifacts might incorrectly favor the latter).
>
> Instead, we argue that the **re-rendered evaluation is a superior metric** for this generative task. This evaluation works as follows: we take the 4-view output from each model, lift it to a 3D representation using LGM (Gaussian splatting), and render novel views from this reconstruction. We then measure the quality of these re-rendered views.
>
> The 2D mode trains the model to handle real-world complexities—occlusions, background clutter, non-canonical poses, and multiple objects—that are absent from the mostly canonical-posed 3D synthetic data. Without this training, the model fails to properly separate object features from environmental context, leading to artifacts copied from retrieval image backgrounds (e.g., strings attached to objects, distorted shapes). If the model fails to separate the object from the background, the background artifacts are 'baked' into the 3D geometry. When rendered from novel views, these artifacts degrade visual quality significantly.
>
> The re-rendered metrics capture this degradation: The model without 2D mode shows inferior CLIP (72.995 vs. 74.278) and FID (82.0468 vs. 80.5406) scores, indicating that background artifacts affect both semantic alignment and overall generation quality, caused by additional artifacts.
>
> Importantly, direct segmentation-based evaluation would not capture this phenomenon effectively, as the artifacts appear as part of the foreground object rather than as separable background elements. The re-rendering evaluation is more appropriate because it measures whether the model can produce outputs that yield coherent, artifact-free 3D reconstructions—which is precisely the goal of our multi-view generation task.
>
> We believe this quantitative analysis, combined with our qualitative ablations in Figure 7, provides comprehensive evidence for the complementary and distinct roles of both training modes.

---

> > ### Author Response · Authors · 2025-11-23
> >
> > **[part 2/3]**
> >
> > **Category Distribution Analysis:** categorization of all 196 OOD-Eval concepts yields the following distribution:
> >
> > | Category  | Percentage |
> > |-----------|------------|
> > | Animals   | 81.1%      |
> > | Vehicles  | 12.2%      |
> > | Food      | 5.1%       |
> > | Artifacts | 1.0%       |
> > | Fungus    | 0.5%       |
> >
> > As noted, Animals and Vehicles are the most represented categories. This distribution is intentional and reflects the nature of **real-world fine-grained named entities** in the long tail. Biological species and vehicle models possess deep, specific subcategories (e.g., "Axolotl", "1998 Fiat Multipla") that allow for precise text-based querying of specific, rare concepts. In contrast, categories like furniture are often defined by attributes rather than unique names, making them less suitable for testing specific concept retrieval.
> >
> > Furthermore, we emphasize that 'Animal' is a broad super-category encompassing immense geometric diversity (e.g., the structure of a Jellyfish vs. a Dog vs. a Bee is far more distinct than the variance within 'Furniture'). Thus, the high percentage of animals does not imply low geometric diversity. Even within these categories, the selected concepts are rare subclasses where high-quality multi-view training data is scarce.
> >
> > Importantly, this does **not** imply that retrieval becomes easier. The concepts in OOD-Eval are rare subclasses, their scarcity, not their parent category, governs retrieval difficulty. Indeed, many of the rare animal or vehicle species in OOD-Eval have few or no high-quality multi-view examples, even in large-scale datasets.
> >
> > To further mitigate concerns about category bias, we highlight in Figs. 6 and 15 (main paper + appendix) the performance of MV-RAG specifically on underrepresented categories, where it continues to outperform baselines, suggesting that the method’s gains do not rely on category frequency.
> >
> > **Quantifying Out-of-Distribution Distance:** To rigorously assess the claim that OOD-Eval prompts are “far from” training data, we conducted a large-scale lexical distance analysis against LAION.
> > We sampled **100,000** LAION captions as a reference corpus.
> >
> >
> > For each of the 196 OOD-Eval prompts, we computed its nearest neighbor in LAION using BM25 and CLIP similarity.
> >
> >
> > As a control, we sampled **2,000 additional LAION prompts** (distinct from the first LAION group) and computed their nearest neighbor similarity within the same LAION corpus.
> >
> > **Results:**
> >
> > | Set Comparison      | BM25 ↑    | CLIP ↑    |
> > |---------------------|-----------|-----------|
> > | OOD-Eval → LAION         | 14.7374   | 73.1238   |
> > | LAION → LAION       | 27.9488   | 80.0988   |
> >
> > We further present the similarities segmented per-category in OOD-Eval:
> >
> > | Category  | BM25 ↑    | CLIP ↑    |
> > |-----------|-----------|-----------|
> > | Animals   | 14.5762   | 72.1227   |
> > | Vehicles  | 16.9395   | 78.3750   |
> > | Food      | 13.1471   | 79.1578   |
> > | Artifacts | 12.2718   | 70.6551   |
> > | Fungus    | 14.5085   | 56.1707   |
> >
> > These results show a large and statistically meaningful separation between OOD-Eval prompts and the LAION text domain. OOD-Eval prompts have **~47% lower lexical similarity (BM25)** and **~9% lower semantic similarity (CLIP)** to LAION compared to LAION captions among themselves.
> >
> > Notably, the BM25 gap is substantially larger than the CLIP gap. This aligns with our findings in Appendix A.7 and Table 7, where we demonstrate that semantic representation models like CLIP lack fine-grained semantic understanding of OOD objects, leading to artificially inflated similarity scores for out-of-distribution concepts. In contrast, BM25 shows superior precision for OOD objects (Table 7), making it a more reliable indicator of true distributional distance for rare concepts.

---

> > > ### Author Response · Authors · 2025-11-23
> > >
> > > **[part 3/3]**
> > >
> > > **End-to-end Inference Latency:** We measured the end-to-end inference time for all baselines on the OOD-Eval benchmark, averaging over multiple prompts. Retrieval was performed using the Rank-BM25 library over the combined test set of COCO + OOD-Eval (54K image-caption pairs), consistent with our experimental setup.
> > >
> > > **Timing breakdown:**
> > > * Average generation time: 6.258 seconds
> > > * Average retrieval time: 0.038 seconds
> > > * **Total inference time: 6.296 seconds**
> > >
> > > As shown, retrieval introduces negligible overhead (~0.6% of total time), and the approach remains highly scalable to larger corpora using established retrieval frameworks (e.g., Pyserini).
> > >
> > > The increase in generation time relative to the base model (MVDream) is attributable to the additional retrieval-attention operations over multiple retrieved image tokens (~4 images per prompt in our experiments). Despite this, inference remains efficient due to our lightweight adapter-based design. These results are now included in the revised manuscript (Table 8).
> > >
> > >
> > > | Model          | Time (seconds) |
> > > |----------------|----------------|
> > > | MVDream        | 1.081          |
> > > | ImageDream     | 1.470          |
> > > | MV-Adapter     | 6.562          |
> > > | SPAD           | 10.250         |
> > > | Trellis        | 12.098         |
> > > | Era3D          | 12.640         |
> > > | MVDreambooth   | 170.411        |
> > > | MV-RAG         | 6.296          |
> > >
> > > To summarize, we sincerely appreciate the thoughtful, detailed, and constructive feedback provided by Reviewer fdtJ. The review offers a deep and nuanced understanding of our method, highlighting both its strengths and the areas where further clarification strengthens the manuscript. We believe that our responses and additional analyses directly address the reviewer’s concerns and we thank the reviewer for the time and care invested in this evaluation. We remain available for any additional questions or suggestions the reviewer may have.

---

### Official Review · Reviewer_p22R · 2025-11-02

**Soundness:** 4
**Presentation:** 3
**Contribution:** 3
**Rating:** 6
**Confidence:** 4

**Summary:**

This work proposes MV-RAG, a text-to-3D generation pipeline that addresses the possible failure of existing methods on out-of-domain (OOD) or rare concepts by first retrieving relevant 2D images from a large database, and then conditioning a multiview diffusion model on these images to generate consistent multiview outputs. The approach uses a hybrid 2-mode training strategy combining structured multiview data with augmented conditioning views and diverse 2D image collections with a held-out view prediction objective to infer 3D consistency, along with a prior-guided fusion mechanism that balances retrieval signals with the model's prior.

**Strengths:**

- The paper is very well written, organized, and easy to follow.
- The paper is well-motivated, tackling an important problem of OOD generation or "rare" concepts that were not sufficiently trained to diffusion models, therefore yielding suboptimal training results when applied to 3D generation, which is a practical and prevalent problem in multiview diffusion models.
- The paper proposes a coherent and intuitive methodology that fits well to their problem at hand: they start from the MVDream architecture, which introduced multiview consistency, and propose to finetune it to accept extended attention from retrieved 2D images for highly detailed, prompt-adhering generation. Their 2-step approach of training a model at a multiview, 4-view generating setting (3D attention) with augmented retrieved images at random viewpoints, and a subsequent 2D image generation setting generating a prompt-adhering image solely from retrieved in-the-wild images makes sense and is well-designed, as well as being cost-efficient training-wise.
- The experimental results show effective performance and are well-organized, demonstrating the effectiveness of this method.

**Weaknesses:**

- The paper could benefit from additional explanation on how it conducts semantic/geometric augmentation at a 3D training setting, and additional details regarding how much augmentation the model can take. For example, if the semantically augmented retrieval images deviate too much from the source 3D asset, the model training may suffer degradation rather than learning 3D consistency: please elaborate on this aspect of the method.
- One question that I have with this method is that 2D image generation is trained to generate the ground truth target image from retrieved images, relying on nothing but the text prompt embedding. However, in many cases, the text prompt may be insufficient in describing the target image, and the model may have difficulty generating the target image with a lack of any geometric / image structural guidance, or may resort to shortcuts such as simple memorization. This problem may be exacerbated as the model training is basically fine-tuning and not available for training on as large-scale data as the original model. Can the authors elaborate further on this issue?

**Questions:**

Please see the Weaknesses section.

---

> ### Author Response · Authors · 2025-11-21
>
> We thank the reviewer for the thoughtful and positive evaluation of our work. We appreciate the recognition of the practical importance of handling out-of-domain concepts in multiview diffusion, as well as the clarity, coherence, and efficiency of our two-stage training pipeline. We also value the acknowledgment of our experimental results. Below, we address the reviewer’s remaining questions and concerns:
>
> **Augmentations**: Thank you for raising this important point. Indeed, if the semantic or geometric augmentations deviate too strongly from the underlying 3D asset, the retrieved views may no longer carry meaningful information about the object’s appearance or structure, which could degrade training. Conversely, if augmentations are too mild, the model would not learn to handle the natural variability present in real-world images.
>
> To balance these effects, we intentionally use moderate augmentations that reflect realistic image variability, such as illumination changes, mild cropping, color jitter, and limited perspective distortion. These transformations preserve the global shape and distinctive features of the object while still providing sufficient diversity for the model to learn robustness and 3D consistency.
>
> We now include an expanded visualization of the applied augmentations in **Fig. 11** and provide implementation details in **Appendix A.6.1**
>
> **Text prompt reliance**: We agree that the text prompt in 2D mode does not fully describe the specific target image. This is intentional: the prompt serves only to anchor the semantic object, while the retrieval images supply the fine-grained visual and structural cues needed to reconstruct the target.
>
> Although the retrievals are not identical to the target, they typically share consistent object appearance, textures, poses, and characteristic scene patterns. These shared properties provide the detailed geometric and visual information that the text alone cannot express. During both training and inference, the model learns to extract these cues from the retrieval set and use them to guide the generation of the target image.
>
> The goal of the 2D mode is therefore not to make the text uniquely specify the image, but to teach the model how to interpret the retrieval images.
>
> To summarize, we appreciate the thoughtful and overall positive feedback from reviewer p22R. We believe our answers address the points raised and further validate the contributions of our work. We will be happy to address any further concerns the reviewer may have.

---

### Author Response · Authors · 2025-12-02
**Summary of Reviewer Discussion and Consensus**

[part 1/2]

We sincerely thank the Area Chair and all reviewers for their thorough evaluation of our work and their constructive feedback. We greatly appreciate the time and effort invested in reviewing our submission, and their detailed feedback has strengthened our paper.

## 1.  Consensus on Strengths

-   **Novelty and importance of retrieval-augmented multiview diffusion**: All reviewers unanimously acknowledged the novelty and significance of applying retrieval-augmented generation to multiview diffusion. Reviewer **fdtJ** found that “_the work has a significant impact_”, reviewer **p22R** found the approach is "_well-motivated, tackling an important problem_", reviewer **UQp6** described it as "_novel and well-motivated_," reviewer **PLGn** found the contribution "_interesting_".

-   **Technical soundness of hybrid 2D-3D training**: The hybrid training scheme received strong praise for its technical design. Reviewer **p22R** characterized it as a "_coherent and intuitive methodology_," reviewer **fdtJ** noted its "_systematic design_," and reviewer **UQp6** found it "_technically sound_".

-   **Experimental rigor and comprehensiveness**: Reviewer **fdtJ** particularly emphasized our "_exceptionally comprehensive and multi-faceted experimental evaluation_" with a "_thorough and systematic_" approach, while reviewer **p22R** praised the "_effective performance and well-organized_" results, and reviewer **UQp6** recognized our "_extensive ablation studies_".

-   **Strong performance on out-of-distribution concepts**: Reviewers **p22R**, **fdtJ**, and **PLGn** all acknowledged the method's effectiveness on OOD concepts, with reviewer PLGn specifically noting "_promising performance on OOD examples_".

-   **Clarity and presentation quality**: The paper's presentation was consistently praised. Reviewer **p22R** found it "_very well written, organized, and easy to follow_," reviewer **fdtJ** described it as "_well-written and structured_" with "_strong dedication to reproducibility_," and reviewer **UQp6** called it "_thorough and well-organized_".

---

### Author Response · Authors · 2025-12-02

[part 2/2]

## 2.  Detailed Resolution of Concerns

In the rebuttal we addressed all reviewers’ concerns through new experiments and clarifications, now incorporated into the manuscript as summarized below:


### **Reviewer p22R:**

-   **Augmentation strategy**: Clarified our semantic/geometric augmentation approach—we use moderate augmentations that preserve global shape while introducing realistic variability; expanded visualization added in Fig. 11 with implementation details in Appendix A.6.1

-   **Text prompt reliance in 2D mode**: Explained that text prompts intentionally serve only as semantic anchors while retrieval images provide fine-grained visual/structural cues; the 2D mode teaches the model to interpret retrieval signals rather than relying on text alone to specify the target image


### **Reviewer fdtJ:**

-   **Quantifying 2D/3D mode contributions**: Provided quantitative re-rendered evaluation showing degraded performance without 3D mode and without 2D mode, confirming its role in geometric accuracy as well as in performance for in-the-wild setting.

-   **Category bias in OOD-Eval**: Provided full statistical breakdown of our benchmark. Showed the most represented categories result from the nature of real-world fine-grained named entities in the long tail for OOD; demonstrated strong performance even on underrepresented categories (Figs. 6, 15)

-   **Quantifying OOD distance**: Conducted large-scale analysis for our benchmark against 100K LAION samples showing OOD-Eval prompts have ~47% lower BM25 similarity and ~9% lower CLIP similarity compared to LAION-to-LAION baseline, rigorously confirming distributional separation

-   **Inference latency**: Provided end-to-end timing comparison (new Table 8) showing MV-RAG at 6.296s with negligible retrieval overhead (0.038s, ~0.6% of total time), remaining efficient and scalable


### **Reviewer UQp6:**

-   **Quantitative improvements**: Clarified PSNR measures in-domain performance (where we match/exceed baselines), while our key contribution is state-of-the-art OOD performance.

-   **Additional baselines**: Wonder3D and SyncDreamer were outperformed by our selected baselines in subsequent publications, confirming our comparison captures current state-of-the-art

-   **Retrieval sensitivity**: Our gating mechanism (Appendix A.2) filters irrelevant retrievals and falls back to text-only generation when needed, ensuring robust performance

-   **Scaling experiments**: Demonstrated improvement upgrading from SD1.5 to SD2.1; our lightweight adapter design allows flexible incorporation of stronger backbones without large-scale retraining


### **Reviewer PLGn:**

-   **Motivation and generalization**: Clarified that MV-RAG learns geometric transfer mechanisms rather than concept memorization, leveraging non-parametric retrieval memory to handle the infinite long tail; demonstrated text-to-image-to-multiview pipeline significantly underperforms MV-RAG, validating the retrieval-augmented approach

-   **Fairness of comparison**: Clarified evaluation protocol already gives baselines best-of-4-viewpoint advantage; conducted additional experiment with TRELLIS using canonical front-views from MV-RAG outputs, confirming performance gap stems from OOD limitations rather than viewpoint sensitivity (new Appendix Figure 14)


To summarize, we believe the comprehensive responses and additional experiments provided during the rebuttal period have thoroughly addressed all reviewer concerns.

---

### Meta-Review · Area_Chair_XRUG · 2026-01-06

**Summary:**

This paper received mixed reviews (4, 6, 6, 6). Given that two reviewers provided relatively brief comments, I read the paper carefully. The paper presents an interesting motivation—using retrieval-augmented generation to address the out-of-domain (OOD) problem. This idea is timely and aligns with current topics in LLMs.

However, the method is primarily built upon multi-view diffusion models, which are becoming less popular. The current trend has shifted toward native 3D generation models, such as TRELLIS and Hunyuan3D. These newer approaches solve many limitations associated with multi-view-based methods and have great potential to scale to larger datasets, effectively mitigating the OOD problem. While it is possible that the proposed method could work with modern generative models, it remains uncertain. Therefore, I recommend rejection. However, I would encourage the authors to demonstrate their method's effectiveness on stronger, modern baselines in the future.

**Reviewer Concerns:**

The authors addressed some of concerns, e.g.,
1. why the retrieval is necessary
2. comparison
3. 2d and 3d disentanglement

To support the claims, the authors also showed extensive experiments.

**Reviewer Scores:**

Based on my experience, I guess reviewers would keep the current scores.

---

### Decision · Program_Chairs · 2026-01-26

Reject